# Dual-bionic superwetting gears with liquid directional steering for oil-water separation

Zhuoxing Liu[1,2], Zidong Zhan[1,2], Tao Shen[3], Ning Li[1], Chengqi Zhang[1,3], Cunlong Yu[1,2], Chuxin Li[4], Yifan Si[5] ✉, Lei Jiang [1,2] & Zhichao Dong [1,2] ✉

Developing an effective and sustainable method for separating and purifying oily wastewater is a significant challenge. Conventional separation membrane and sponge systems are limited in their long-term usage due to weak anti-fouling abilities and poor processing capacity for systems with multiple oils. In this study, we present a dual-bionic superwetting gears overflow system with liquid steering abilities, which enables the separation of oil-in-water emulsions into pure phases. This is achieved through the synergistic effect of surface superwettability and complementary topological structures. By applying the surface energy matching principle, water and oil in the mixture rapidly and continuously spread on preferential gear surfaces, forming distinct liquid films that repel each other. The topological structures of the gears facilitate the overflow and rapid transfer of the liquid films, resulting in a high separation flux with the assistance of rotational motion. Importantly, this separation model mitigates the decrease in separation flux caused by fouling and maintains a consistently high separation efficiency for multiple oils with varying densities and surface tensions.

Safe drinking water is recognized as a basic human right to sustain healthy livelihoods and is fundamental to maintaining the dignity of all human beings. With population expansion and environmental destruction, 1.8 billion people are facing absolute water shortage and 4 billion people (half of the world population) are experiencing water scarcity[1]. Especially, 1 billion Indians, 300 million Chinese[2], 130 million Bengalese, 120 million Pakistani (85% are in the Indus basin), 110 million Nigerian, and 90 million Mexicans are facing severe water scarcity during at least part of the year[1], and 63 million Americans (almost one-fifth of the population) are exposed to water contamination, which is higher than any situation during recent decades[3]. Purifying oily wastewater caused by oil spills, industrial chemical leaks, and sanitary sewage into clean water is essential to address various social and economic inequities[4,5].

Among oily wastewater, oil spills have disastrous consequences for local ecosystems and can be expensive due to the loss of oil and the costs involved in the clean-up[6,7]. Figure 1a summarizes the oil spills from 1970 to 2022, where spills always occur in economically developed areas with large populations[8]. Oil spills are disasters that can have severe social, economic, and environmental impacts. According to the World Bank, if not correctly treated the oily wastewater pollutions, the GDP growth rates could decline by 6% with a loss of $4.5 trillion by 2050[9]. Moreover, although existing advanced materials or systems, such as superwetting membranes and 3D porous bulks, demonstrate the potential to solve these global and frontier challenges than traditional materials[10–13], these materials or systems still face some inevitable deficiencies: 1. Weak antifouling ability resulting in decreased separation flux;[14] 2. Lack of processing capacity to handle light/heavy

[1]CAS Key Laboratory of Bio-inspired Materials and Interfacial Science, Technical Institute of Physics and Chemistry, Chinese Academy of Sciences, 100190 Beijing, China. [2]School of Future Technology, University of Chinese Academy of Sciences, 100049 Beijing, China. [3]Key Laboratory of Bio-Inspired Smart Interfacial Science and Technology of Ministry of Education, School of Chemistry, Beihang University, 100191 Beijing, China. [4]Suzhou Institute for Advanced Research, University of Science and Technology of China, Suzhou 215123 Jiangsu, China. [5]Department of Biomedical Engineering, City University of Hong Kong, 999077 Hong Kong SAR, China. ✉e-mail: yifansi@cityu.edu.hk; dongzhichao@mail.ipc.ac.cn

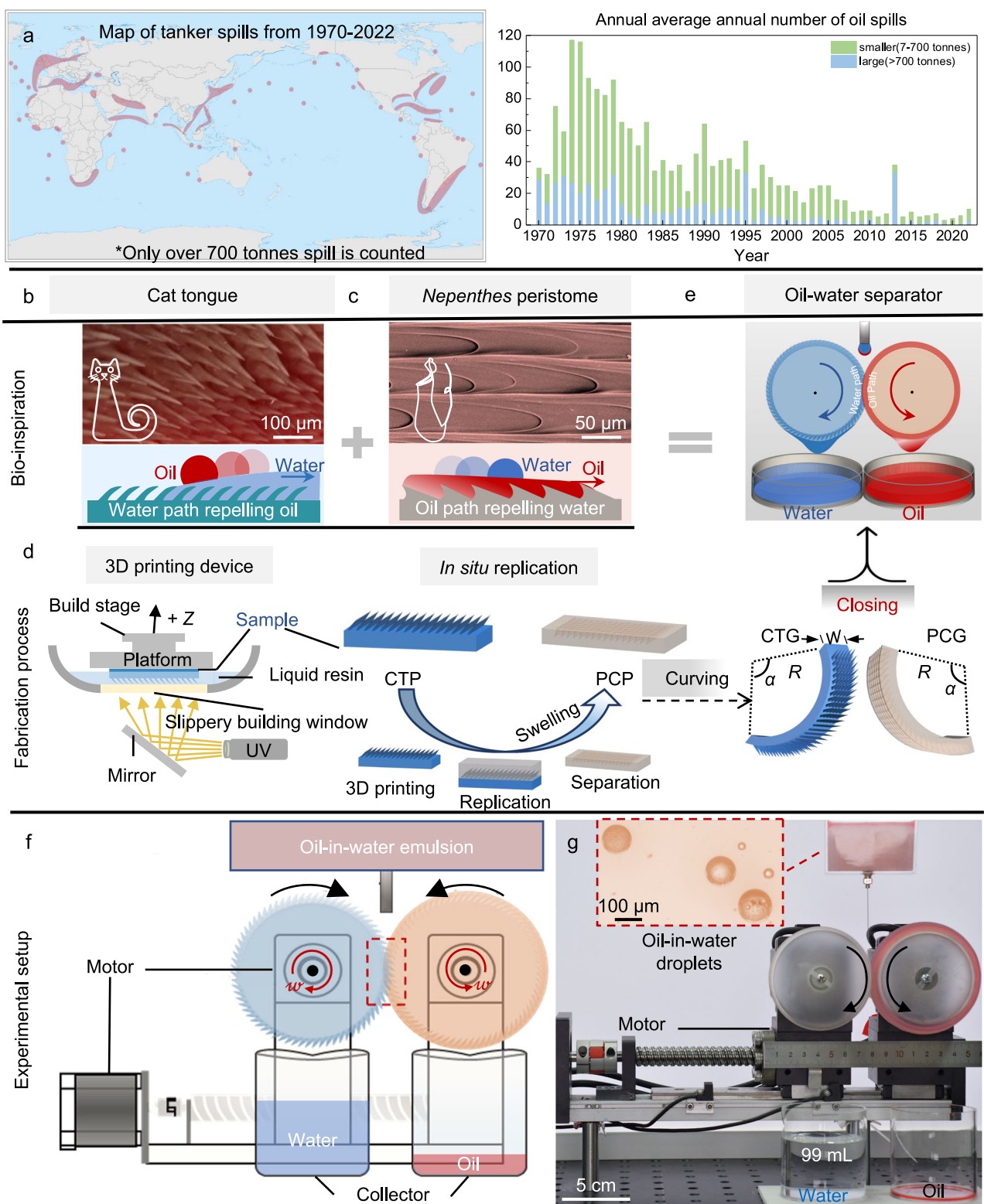

**Fig. 1 | Designing dual-bionic gears for oil-water separation. a** Map of tanker spills that spill size over 700 tonnes is counted from 1970 to 2022, and the statistics for the number of tankers spills >7 tonnes from 1970 to 2022. **b** and **c** Sketch portrays the surface structural design learned from the cat tongue and *Nepenthes* peristome. **d** Scheme of the fabrication process. Curving and closing the dual-bionic substrates achieve the designing gears. **e** Schematic diagram of oil-water micro-drop separation process via rotating dual-bionic superwetting gears. **f** Draft design of the experimental setup. **g** Photograph of the outcome of the oil-in-water emulsion separation. Inset, the oil-in-water emulsion contains microscale oil droplets in the water phase. Key components include the platform holder, rotation motor, superwetting gears, tank, and collector. Two gears are complementary with each other with opposite rotation directions by the same angular velocity. Source data for a are provided as a Source Data file. Image (**b**) reproduced with permission from ref. 29, Proc. Natl. Acad. Sci. U.S.A.

oils coexistence in one system;[15–17] 3. Low feasibility for multi-scale oil-water systems with different densities, especially for oil-water micro-drops and oil-in-water emulsions[17,18]. Scalable oil-water separation devices have progressed in recent years, including functionalized trawling nets[19], large separators-skimmers[20], floating wells[21], etc. But the underlying mechanism of these devices is also based on super-wetting membranes or sponges[20–23]. Although the total amount or separation rate has been improved, the limitations of the closed and static separation modes are not broken. The above three deficiencies still exist. Besides widely researched oil-spill cleaning-up methods[10–13], the oil-water separation for sanitary sewage treatment, another important environmental and healthcare problem[15], is rarely discussed. It is, therefore, a dramatically urgent task to build a facile and effective oil-water separation strategy for multi-scale oil-water micro-droplets.

Nature's inspirations in their structures and profound principles provoke new design blueprints for the applications of functional surfaces[24–32]. Cat uses hollow spiky to wick saliva into fur[29], and the *Nepenthes* pitcher peristome uses arrayed cavity structures transporting lubricant directionally to construct a liquid-repellent slippery surface for capturing insects[24–27]. If adjusting to the same scale, the combination of the cat-tongue and *Nepenthes*-pitcher-peristome cavities could form complementary topological structures while retaining their respective superwetting properties. As a hypothesis, designing a dual-bionic system will contribute to a feasible strategy to solve current challenges in oil-water separations.

Inspired by the cat tongue and *Nepenthes* pitcher peristome, we demonstrate a dual-bionic superwetting gears overflow strategy via 3D printing to achieve rapid, continuous, lossless, and efficient separations of oil-water micro-drops in multi-scales and oil-in-water emulsions with varying-densities utilizing the synergies of superwettability, complementary topology morphology, and mechanical motion. The designed dual-bionic model with millimeter-scaled and micro-scaled structures steers liquid overflowing, spreading and separation (Supplementary Movie 1). This facile and scale-up 3D printing strategy with a low carbon footprint displays excellent antifouling ability and high separation efficiency. The extrusion force from complementary topological structures provides a novel demulsification way with high efficiency of ~99%. The decrease in separation flux caused by fouling does not appear even after 600 min of separation.

## Results

### Designed dual-bionic gears for oil-water separation
The central concepts for our dual-bionic separation strategy are derived from a combination of the spiky cat tongue[29] (Fig. 1b) and slippery liquid-infused *Nepenthes* pitcher peristome[24–26,30] (Fig. 1c). A 3D printing method[33] is used to construct the bio-mimetic cat tongue surface, and a replicating method is used to fabricate peristome-mimetic surface (Fig. 1d). The imaging unit is projected through a building window to photo-cure the liquid resin that is deposited on the platform's bottom. Our fabrication method advances commercial 3D printing techniques by constructing a slippery lubricant interface for fast and continuous printing. Using the existing 3D printing technique to construct flexible resin architectures, researchers have found that the adhesion at the curing interface can lead to printing failure and influence printing resolution[34]. As an advantage, in our experiment, slippery lubricant[30,33] is deposited onto the building window to reduce adhesion between the cured resin and the curing interface and reduce the heat accumulated at the interface. After the 3D printing process, we then modified the biomimetic cat tongue plane (CTP), cat tongue mimetic photocurable resin teeth, with a coating of $TiO_2$ nanoparticles to enhance water affinity (Supplementary Fig. 1).

Replicating the surface morphologies of the CTP by poly-dimethylsiloxane (PDMS), we get the peristome-inspired cavity plane (PCP) (Supplementary Fig. 1e). The immersing of PDMS into the oil phase can swell the structures with a swelling ratio of 101–105% for chili oil, salad oil, FC-72 and silicon oil-100, ~110% for silicon oil-20 and *n*-hexadecane, 116% for tetrachloromethane, and 135% for *n*-hexane, respectively (Supplementary Fig. 2a). The FTIR spectra showed that oil swelled into PDMS. And the infrared spectrum of PDMS immersed with other oils is shown in Supplementary Fig. 2b, c. The final size of PCP was therefore balanced by the design of the replicating model and the total swelling ratio of the oil-infused PDMS surface. The curving and combination of the designed and dual-biomimetic substrates form the oil-water separator (Fig. 1e).

Based on the static dual-curved oil-water microdroplet separation device we reported earlier[13], the upgraded dynamic dual-bionic superwetting gears device has been successfully designed. Figure 1f shows the schematic diagram of the experimental device that consists of five key components: platform holder, rotation motor, superwetting gears, tank, and collector. Two gears are complementary with each other with opposite rotation directions by an angular velocity $\omega$ of 6 rpm. The oil-in-water emulsion is a supersonic mixture of the oil phase, red-dyed *n*-hexadecane ($0.77\,g\,cm^{-3}$), and the water phase. Gears are pre-wetted by water (Supplementary Fig. 2d) and corresponding oil (Supplementary Fig. 2b) to form superlyophilicity and slippery properties. When the oil-in-water emulsions, oil microdroplets in the water phase, fall in the intermediate junction of the gears, the oil and water phases show different overflow-spreading behaviors on their surfaces, creating independent oil and water phases and dripping into the collectors after accumulation. As Fig. 1g reveals, when the oil-in-water emulsions are deposited on gears at a flux of $30\,\mu L\,s^{-1}$, water can spread quickly on the cat tongue gear (CTG) surface, and oil can spread on the peristome-inspired cavity gear (PCG) to form two liquid films simultaneously on the two sides, respectively. The separation efficiency, $\eta = (V_c / V_d) \times 100\%$, is approaching around 99% for *n*-hexadecane-water mixture, where $V_c$ and $V_d$ are the volumes of water or oil that are collected and deposited. After 1 h, ~99 mL of clean water and ~1 mL of red-dyed *n*-hexadecane are collected (Fig. 1g, Supplementary Movie 2). Moreover, we found that after a 1 h separation experiment, no obvious reduction or damage was observed on the super-hydrophilic resin gear or PDMS gear (Supplementary Fig. 1d and f).

### Screening dual-bionic gears designed for sustained oil-water separation
Figure 2a–c shows the detailed surface structures and wettabilities. The biomimetic cat tongue plane (CTP) is shown to consist of periodically arranged teeth (Fig. 2a, Left). The height ($h$), pitch ($p$), length ($l$), and tilting angle ($\alpha$) of teeth are 1 mm, 1 mm, 1.15 mm, and 60°, respectively (Scheme of Fig. 2a, Left). The teeth exhibit dual curvatures with a transverse curvature of radius $r_1$ of 0.1 mm and a longitudinal curvature of radius $r_2$ of 5.0 mm (Supplementary Fig. 1a and c). Such a tooth enables water wicking both in and out of the surface plane (Supplementary Fig. 1b). CTP exhibits superhydrophilicity with a water contact angle of ~0° and an underwater oil contact angle of ~155° (Fig. 2b). Continuous deposition of water can achieve unidirectional transport along the titled direction, and continuous deposition of oil can achieve transport along the arrayed cavity structures directionally (Supplementary Fig. 3). After oil swelling, PCP shows super-oleophilicity and slippery behavior with an oil contact angle of ~0° and an under-oil water contact angle of ~152° (Fig. 2b and Supplementary Table 1).

The different superwettability of two biomimetic substrates is one of the fundamental factors for water-oil separation and demulsification. The biomimetic cat tongue plane (CTP) exhibits extremely low adhesion to various oil drops under a water environment (red plots in Fig. 2c). And the peristome-inspired cavity PDMS plane (PCP) exhibits extremely low adhesion to water drops under various oil environments (blue plots in Fig. 2c). This allows the dispersed phases to spread and adhere only on one surface of the gears.

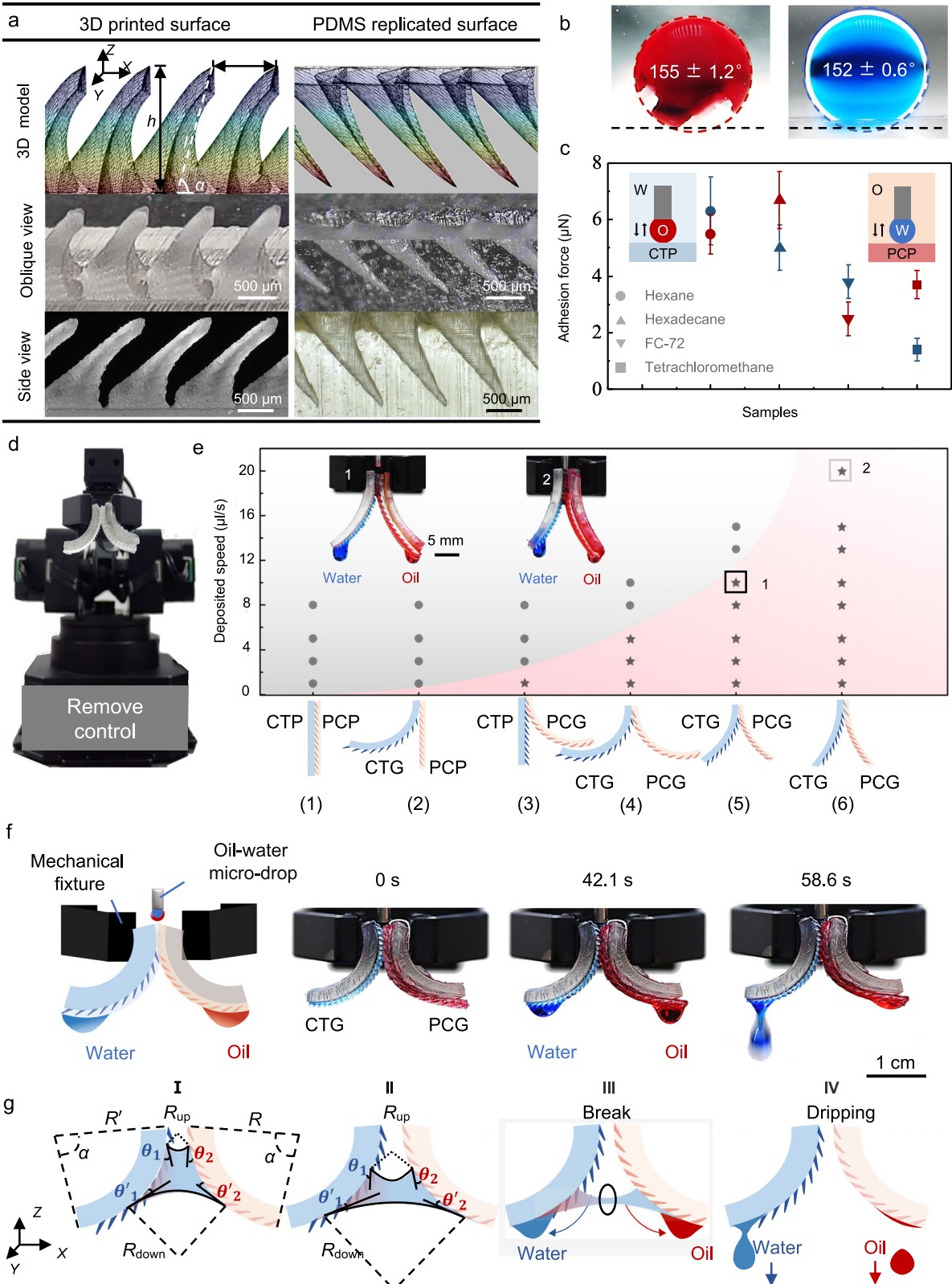

Next, we put forward a progressive dual-bionic strategy from 2D plane to 3D gear. A detailed experiment is performed to screen the optimal parameters for effective oil/water separation (Fig. 2d–g and Supplementary Fig. 4). Our motivation is to utilize the synergies of superwettability (to realize the overspread of different liquids on different surfaces and guide the liquids' directional transport), centrifugal force (to accelerate liquids overflow, promote the liquid bridge

breaking) and extrusion force (to break the dispersed phase of the emulsion) synergies to enable rapid, continuous, and efficient oil-water mixture and oil-in-water emulsion separation. Based on this motivation, a mechanical arm is used to achieve remote control of the oil-water micro-drop separation process and record the separation amount (Fig. 2d). We can locate oil-water leakage places by a software control system. The bio-mimetic 2D planes can be curved into a 3D arc

**Fig. 2 | Screening dual-bionic gears designed for sustained oil-water separation.** **a** Surface morphologies of the 3D printed cat tongue plane (CTP) and replicated PDMS peristome-inspired cavity plane (PCP). **b** Underwater hexadecane contact angle on the CTP (Left) and under-hexadecane water contact angle on the PCP (Right). **c** Liquid adhesion force of water drop on the CTP under different oil samples and oil samples droplets on the PCP underwater. **d** Experimental setup of the oil-water separation device. **e** Phase diagram of effective oil-water separation in the relation between the liquid deposited speed and the sample combinations. Sketch (bottom) reveals the sample combinations: (1) biomimetic cat tongue plane (CTP) + peristome-inspired cavity plane (PCP), (2) biomimetic cat tongue gear (CTG) + PCP, (3) CTP + PDMS cavity gear (PCG), (4) CTG + PCG (R = 12 mm), (5) CTG + PCG (R = 24 mm). (6) CTG + PCG (R = 36 mm). **f** Scheme of the experimental setup for the mechanical-fixed oil-water separator (Left). Time sequences of the separation process (Right). **g** Sketch portrays the air-water-oil interface topography inside dual bionic gears. Data in (**c**) is shown as mean ± SD and the error bar represents SD. Source data for (**c** and **e**) are provided as a Source Data file.

shape by a 3D printed model (Fig. 2e). The arc of the photo-cured cat tongue substrate and the peristome-inspired cavity substrate, which we termed CTG and PCG, have the same radius of curvature ($R$ ranging from 12 mm to 36 mm), radian ($\alpha$ ranging from $\pi/3$ to $\pi$), and width ($W$ of 1 mm).

Mechanical fixture clamps together the top of biomimetic substrates to enable occlusion between the superhydrophilic teeth[35] and superoleophilic cavities. Through software control, our samples can be positioned in place. As schematically shown in Fig. 2e, we have tested six groups of photocurable resin and PDMS samples with different $R$ and $\alpha$: (1) biomimetic cat tongue plane (CTP) + peristome-inspired cavity plane (PCP), (2) biomimetic cat tongue gear (CTG) ($R = 12$ mm, $\alpha = \pi/2$) + PCP, (3) CTP + PDMS cavity gear (PCG) ($R = 12$ mm, $\alpha = \pi/2$), (4) CTG ($R = 12$ mm, $\alpha = \pi/2$) + PCG ($R = 12$ mm, $\alpha = \pi/2$), (5) CTG + PCG ($R = 24$ mm, $\alpha = \pi/2$), (6) CTG + PCG ($R = 36$ mm, $\alpha = \pi/3$). The initial deposited rate, $v$, is set at the same value for water or oil flow with $v$ of 1–20 $\mu$L s$^{-1}$. So, the overall deposited speed is $2v$, in the 2–40 $\mu$L s$^{-1}$ range.

A phase diagram of the relation between liquid deposited speed and six group combinations is mapped in Fig. 2e. The experimental results can be divided into two situations: the gray-filled circle indicates experiments in which oil-water micro-drops fail to be separated, and the pink-filled star means oil-water micro-drops can be separated. In group (1), oil and water have a trend of spreading separately between the two samples' cracks. However, oil and water meet at the bottom exit and recombine into a micro-drop again. In group (2), oil and water tend to spread on the curving CTG together. Unexpectedly, the combination of CTP and PCG can achieve the desired separation of oil-water micro-drops. Oil extends along the PCG, rapidly forming a hydrophobic slippery surface. Then, water can spread on CTP. Groups (1) and (2) cannot achieve oil-water micro-drops separation. In group (3), micro-drops can be separated under a low rate of $v = 1 \mu$L s$^{-1}$; however, the separation will fail when $v$ increases to 3 $\mu$L s$^{-1}$. Group (4) with two arcuate samples can perform successful separation under the maximum deposited speed of 5 $\mu$L s$^{-1}$. Surprisingly, the maximum deposited speed increases along with the increase of $R$. As for group (5), $R = 24$ mm, the maximum deposited speed can be as high as 10 $\mu$L s$^{-1}$. The separation does not fail even when the maximum deposited speed increases beyond 20 $\mu$L s$^{-1}$ by group (6).

Time-lapse images of the oil-water separation process are shown in Fig. 2f, and the complementary scheme is shown in Fig. 2g. Oil-water micro-drops are generated and deposited onto the occlusion place. Water and oil spread along the curving CTG and PCG, respectively. Water drops and oil drops appear on the bottoms at the same time. With oil-water micro-drops constantly deposited, water and oil drops will fall under gravity. The detailed separation mechanism can be found in the supplementary text of the method part. Although the device displays superior oil-water micro-drops separation performance, the liquid film permanently submerged into micro-nanoscale structures of the static superwetting surfaces during the separation process and then lost efficiency[13,15,34,36]. To overcome this challenge, we further close the arc, namely, $\alpha = 2\pi$, to obtain the wheel-like biomimetic cat tongue gear (CTG) and a peristome-inspired cavity gear (PCG). Topology bionic features are useful and necessary for oil-water

separation, we also performed the control experiment with two gears without bionic features. Even if the two gears have the same super-wettabilities as the bionic gears, they will still lose the ability to separate oil and water (Supplementary Fig. 5).

## Separator design and working mechanism

A draft design can adjust the rotation and occlusion of the gears to achieve oil-water separation (Fig. 3a). The rotation motor that along the platform holder can adjust the occlusion between the superwetting gears (Fig. 3b). The detailed parameters of the experimental device were characterized by the X-ray imaging process and shown in Fig. 3c. Two gears with $R$ of 36 mm that are equipped with superwetting gears are complementary with each other with opposite rotation directions by the same angular velocity of $\omega$. The occlusion and rotation guarantee the separation process continuously and effectively. As shown in Fig. 3b, oil-in-water droplets can be separated into pure water and oil phase with a separate volume of 50 mL and an efficiency of 99% only after 30 min collection process.

First, the occlusion effect is characterized by the macro-, X-ray imaging (Fig. 3c, inset), and Micro-computed tomography (Micro-CT) (Fig. 3d). The complementary topological structures[33] of gears have played a vital role in the separation process. From macro-imaging process, teeth and cavities match together (Fig. 3c, inset). Moreover, Micro-CT is utilized to gain a 3D view of the internal state of the complementary area in high magnifications[37]. From the oblique view image, each column of tilted teeth is one-to-one correspondence to the cavities. Besides, the cross-sectional views of the 3D model demonstrate the inner tilted teeth and cavities are complementary well without gaps (Fig. 3d).

Next, the role of rotation gears in water-oil microdroplets separation is demonstrated in magnified views (Fig. 3e). The teeth and cavities structures can lead liquid films to directional spread rapidly along the gears to the complementary area of the dual bionic gears. Due to the capillary effect, water can form a concave meniscus between the teeth and cavities to get additional *Laplace* pressure[38], $P = \gamma/R_{up} \cos\theta$, inside the V-shaped curve between neighboring gears for the capillary suction of the oil-water mixture to the complementary part (Fig. 3e), where $\gamma$ and $R_{up}$ are the surface tension and radius of curvature (RoC) of the meniscus, and $\theta$ is the constant angle of the liquid on the gear surface. Besides, gravity persists and keeps the oil-water mixture from entering the complementary part to promote the separation process in a long term. In the microscale, because of the capillary suction[13], oil gathers in the cavity (Fig. 3f). Along the $Z$-axis, $F_z$ of the liquid column can be determined as $1/2\left[\left((\gamma_w + \gamma_o)/R_{up} - (\gamma_w + \gamma_o)/R_{down}\right)S - Vg(\rho_w + \rho_o)\right]$, where $\gamma_w$ and $\rho_w$ are the surface tension and density of water, $\gamma_o$ and $\rho_o$ are oil surface tension and density of oil, $R_{up}$ and $R_{down}$ are RoC of the upper and downward menisci, $S$ is the cross-section area of the complementary part in between two gears, $\rho$ and $V$ are the density and volume of the liquid column, and $g$ is the acceleration of gravity, respectively. The elastic deformation of complementary topological structures detected during the rotation process can tear the oil-water films apart to separate water and oil phases into two sides (Fig. 3g).

The separation threshold occurs with ever-increasing $V$ of oil-water micro drops that are constantly deposited, making the direction

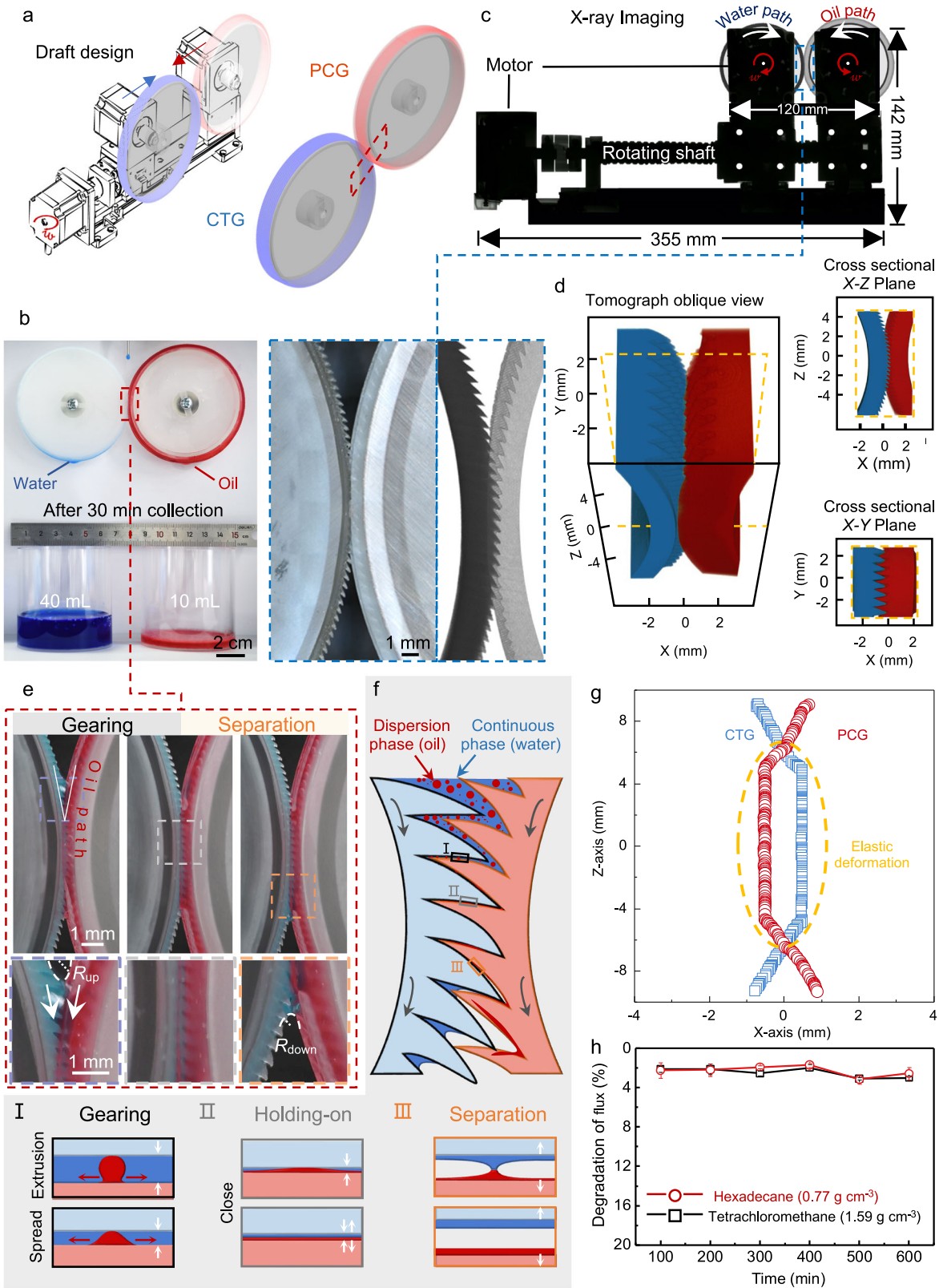

of $F_z$ switch to the -Z-axis resulting in the downflow of the liquid column. Based on the screening experiment shown in the phase graph in Fig. 2e and scheme in Fig. 2g, $R_{up}$ and $R_{down}$ start getting larger simultaneously, but the change rate of $R_{down}$ is much faster. According to the geometrical relationship of force models, a smaller $R$ of the gear leads to a more significant difference between $R_{up}$ and $R_{down}$, causing a greater $F_z$, resulting in a large downward driving flow to spread along

gears. This is why a more significant $R$ we used here (Fig. 3a–c) is more beneficial to oil-water separation.

Under the synergistic effect of these two factors, occlusion effect and rotation, the schematic diagram of the emulsion separation mechanism is shown in Fig. 3f. The continuous phase (water) is dripped with the dispersed phase (oil) to the active region above the contact area. As the gears rotate, complementary topological structures are

**Fig. 3 | Dual bionic separating gears and separation mechanism. a** Exploded view of the draft design for separating gear system. **b** Image of the oil/water separation process. **c** Overlay mapping of X-ray imaging characterizes the device parameters. Inset is the enlarged optical and X-ray images portraying the occlusion effect. Teeth and cavities match together. **d** Micro-CT imaging of the occlusion effect between the teeth and cavities from the oblique and cross-sectional views. The complementary area is the focus part during the three-dimensional reconstruction process. **e** Time sequence images of the water-oil separation process in magnified view. **f** Emulsion separation mechanism with gearing, holding-on, and separation process. Schematic illustration of the O/W emulsions separation by dual-bionic superwetting gears. This process can be divided into 3 regions.

Emulsion micro-drops, as dispersion phase, are going to enter the complementary area driven by gears. During the extrusion process, the dispersion phase would be squeezed and spread out into a liquid film on one of the two gears. **g** Plots of trajectories of the two gears near the complementary area. Time tracking statistics of displacement of gears during the rotating process along the X-axis and Z-axis directions. The complementary area is the focus part. The open blue circle and open red rectangle represent the trajectories of CTG and PCG, respectively. **h** Plots of flux degradation for the separation of hexadecane-water and tetrachloromethane systems. Data in (**h**) is shown as mean ± SD, and the error bar represents SD. Source data for (**g** and **h**) are provided as a Source Data file.

inserted into each other, gradually reducing the liquid space. During the extrusion process, dispersion phases would be squeezed. In the confined space, the dispersed phase is squeezed to contact the two gears surfaces (Fig. 3f, from I to II). However, due to the difference in wettability, the dispersion phase is selected to spread rapidly on one gear's surface, forming a thin liquid film altogether (Fig. 3f, II). The space between the two gears gradually increases in the lower separation region, causing the two films to separate (Fig. 3f, from II to III). The trajectories of the two gears near the complementary area are tracked in Fig. 3g. It can be detected that the PCG presents visible elastic deformation, which is evidence of the extraordinary extrusion force, providing a scraping effect on the water phase. Then, with the breaking of the liquid bridge, the oil and water phases formed independent liquid films on the surface of the two gears. The dispersion phase can grow into pure droplets and be collected at the bottom of the gears. This explains why our method can achieve a large volume of oil-in-water (O/W) emulsion separations without an external field (Fig. 1g). Some comparisons with other studies are shown in Supplementary Table 2. Contributing to the stability of the structural design, the degradation of flux is <3% even after 600 min of operation for both the water-hexadecane mixture and the water-tetrachloromethane mixture (Fig. 3h).

## Dual-bionic gears for sustainable oil-water separation

The spills of crude oil have caused significant marine pollution. Crude oil-in-water emulsions formed by sea wave impact during oil spills are very common but extremely hard to be separated, leading to economic losses and environmental pollution[3,5–7,11,39,40]. Oil-water emulsion separation is one of the most difficult tasks in oily wastewater treatment among multi-scale oil-water micro-drops[39–46]. How to achieve oil-in-water (O/W) emulsion separation by one material without out-field stimulation to transform materials' wettability remains a huge challenge.

Dual-bionic gears achieve the separation of the oil-water mixtures and emulsions in a high efficiency and sustainable way (Fig. 4a). The process is carried out only under gravity at room temperature. The physical parameters $R$ and $W$ of gears are 36 mm and 10 mm optimized by the screening experiments mentioned above, respectively. To mimetic the emulsion by the water wave impact process, we use an ultrasonic disrupter to prepare a supersonic mixture of oil and water phases into the emulsion. The pump sucks up O/W emulsion to the entrance of the separator. The oil and water can be collected from two different exits through the same extrusion and centrifugation process with the assistance of a brush (Fig. 4b). During the process, the brown crude oil-in-water emulsion is opaque before separation. As crude oil is a low-concentration dispersion phase in the mixture, receiving pure oil droplets through a small volume is challenging. However, the dispersion phase in emulsions always is the desired part[34,39,41]. In a massive volume experiment, we can obtain pink n-hexadecane-in-water emulsion via ultrasonic. Oil droplets with tens or hundreds of microns are dispersed in water. In addition, the emulsion dispersing phase's particle size and distribution were analyzed by the laser granularity instrument and the average size of the oil droplets is ~100 μm (Supplementary Fig. 6). In a

magnified stereomicroscope image, we can observe many red micro-scale oil droplets dispersed in water before the separation process. After the separation, the opaque emulsion is separated into clear water and oil phases (Fig. 4b). Ultimately, we could get ~900 mL water and ~7.5 mL oil, respectively (Fig. 4b). The separation rate was set at 40 μL s⁻¹, and the total separation time was about 420 min. Besides, we tested and compared the separation efficiency of n-hexadecane/water emulsion at different rotation speeds within our device capacity (Supplementary Fig. 7). The results showed that the maximum separation efficiency is 99.4% when two gears with opposite rotation directions by an angular velocity $\omega$ of 6 rpm.

In practice, both sophisticated light and heavy oil-water co-existed in industrial or domestic wastewater. Existing superwetting membranes or bulk materials typically show a tremendous restriction which is only to separate one type of light (or heavy) oil-water mixed system[41,47,48]. It is necessary to develop a strategy to handle light and heavy oil-water separation simultaneously. Our dual-bionic separation device can solve this problem effectively (Supplementary Fig. 8). The separation efficiency is around 99% for different oils (Fig. 4c). The lowest surface-tension oil, FC-72, can also be separated. Besides, there are no blocking phenomena caused by density, which is the main essential reason why it can be used to separate varying-density oil-water drops. Taking advantage of the designing dual-bionic gears, we find that the separation efficiencies, $\eta = (V_c/V_d) \times 100\%$[9,14], are all around 99% for both light oil systems, such as n-hexadecane (0.77 g cm⁻³) and heavy oil systems, such as tetrachloromethane (1.59 g cm⁻³) with a high separation rate of 20 μL/s, where $V_c$ and $V_d$ are the volumes of oil collected and oil deposited (Supplementary Fig. 8). The permeating flux also can be calculated by $J = 2V_c/HWt$, where $H$ is the height across the complementary topological area, $W$ is the width of the gear, and $t$ is separation time. When $v$ is 20 μL s⁻¹, the $J$ is approximated to be as high as 2000 L m⁻² h⁻¹.

Weak mechanical strength (or chemical durability) and degradation of separation flux caused by fouling also limit the superwetting membranes for practical applications. To pursue higher flux, according to the Hagen-Poiseuille equation, $J = \frac{\varepsilon \pi r_p^4 \triangle p}{8 \mu L}$, the membrane system needs smaller thickness and greater porosity, which inevitably leads to weak mechanical strength and lower breakthrough pressure, where $J$ is the permeation flux of liquid, $r_p$ is the effective pore radius, $\varepsilon$ is the porosity, $\mu$ is the dynamic viscosity of the liquid, $\triangle p$ is the applied pressure, and $L$ is the thickness of the membrane. Our dual-bionic separator can solve it in a facile method. Because of the synchronous separation mechanism, liquid films on sample surfaces would not be infected by other liquids. Liquid films can be renewed continually by deposited liquid like a deep washing. A long-time separation experiment is also demonstrated to prove great antifouling ability. The separation flux statistics are carried out every 100 min. There is no significant decrease (<5%) in light/heavy oil-water separation flux even after 600 min of separation operation, indicating the excellent antifouling ability of our device (Fig. 3h). Compared with the current membrane and foam materials, our dual bionic separator has superior mechanical strength, antifouling ability, and less fabrication time (Fig. 4d and Supplementary Table 2). In general, our strategy displays

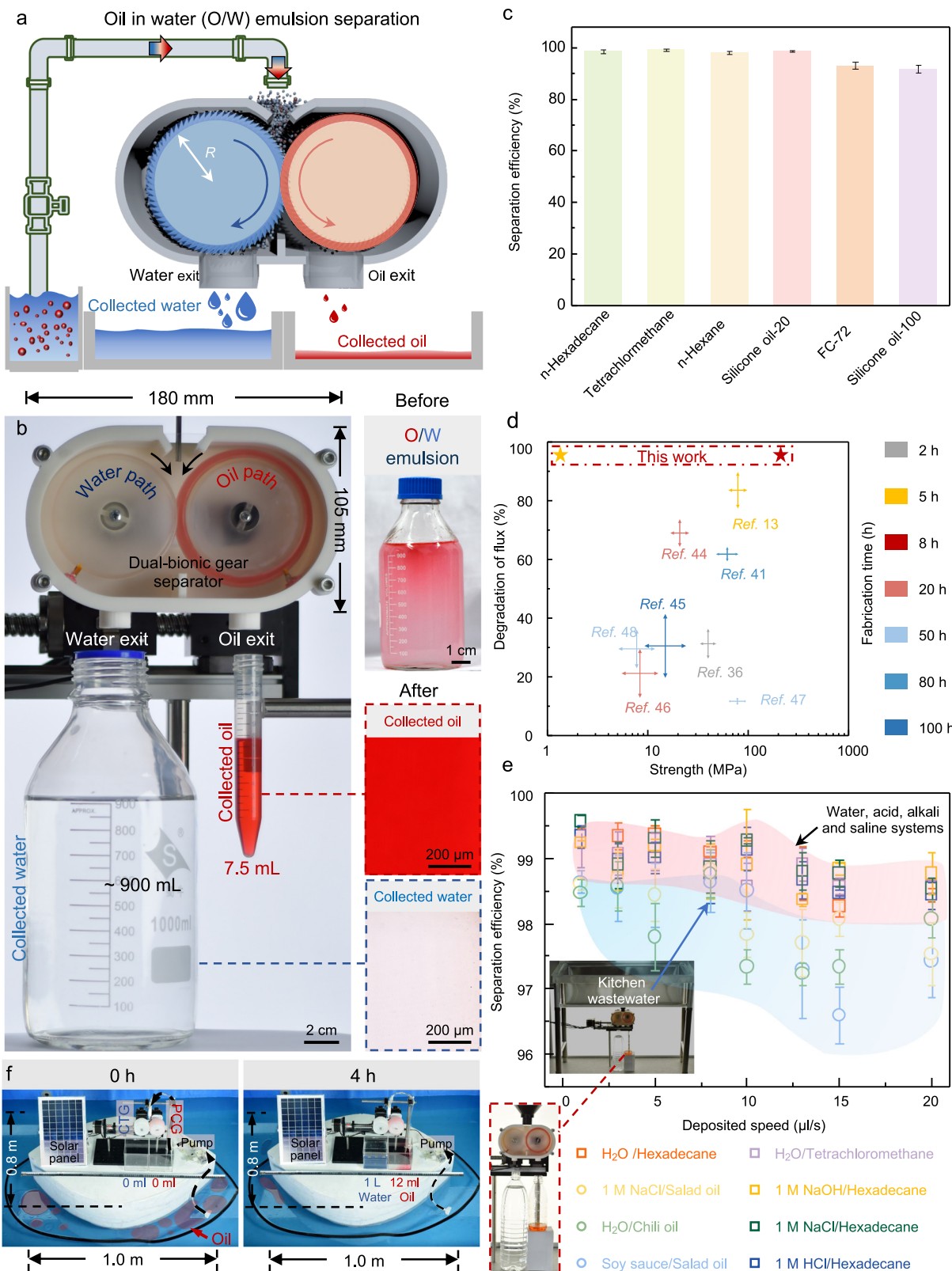

high-performance stability and antifouling ability with more sustained separation abilities.

Besides these industrial oil-water micro-drops, we also realize the separation of kitchen wastewater and conventional complex oil-water mixing systems produced by daily human life[4,49,50]. We simulated three groups of kitchen wastewater micro-drops: 1 M NaCl/salad oil, H₂O/chili oil, and soy sauce/salad oil. In detail, the components of the

simulated kitchen wastewater are more complex with higher viscosity. Still, the separation efficiency could be maintained above 97%, even if the flux is as high as 20 μL/s. Simultaneously, the effects of acid, alkali, and salt systems on separations also have been studied here (Fig. 4e). It has been proved that density, surface tension, and pH value have little effect on separation efficiency, and separation efficiency >98% is easy to achieve. That means our designing dual-bionic gear separator has

**Fig. 4 | Dual-bionic gears for sustainable oil-water separation. a** Schematic of the integrated dual-bionic gears separator for oil-in-water (O/W) emulsion. The pump sucks up O/W emulsion to the entrance of the separator. Through the extrusion and centrifugation process with the assistance of a brush, pure oil or water phase can be collected from two exits. **b** Left: Experimental setup of the dual-bionic gears for sustainable oil-water separation. Right: Optical and stereomicroscopic images of O/W emulsion and collected water and oil. In a magnified stereomicroscope image, the opaque emulsion is separated into clear water and clear oil phases after the separation. **c** Separation efficiencies of various oil-water mixtures with different densities, viscosities, and surface tensions. **d** Plots of flux degradation vs. material strength for comparing our method and other separation systems. **e** Separation efficiencies of different oil-water micro-drops. Inset, the collection of simulated kitchen wastewater. Three types of simulated kitchen wastewater exist: 1 M NaCl/ salad oil, $H_2O$/chili oil, and soy sauce/salad oil. **f** Proof-of-concept experiment of dual-bionic superwetting gears device to separate water and oil in a swimming pool. Data in (**c** and **e**) are shown as mean ± SD, and the error bar represents SD. Source data for (**c**–**e**) are provided as a Source Data file.

considerable potential to be extended to our daily life, bringing about a real revolution in rubbish classification, energy reutilization, and food security.

A proof-of-concept experiment is performed in a swimming pool with a length of 2.2 m, a width of 1.5 m and a water depth of 0.5 m to simulate the cleaning ability in the oil spill accident. The device is mounted on a boat of 1.0 m long and 0.8 m wide, where a pump driven by the solar cell drips contaminated seawater onto the gears (Fig. 4f, Supplementary Fig. 9). The effective ability of our dual-bionic superwetting gears device can ensure that the separation flow does not decrease for 240 min.

## Discussion
In summary, we have presented a dynamic dual-bionic superwetting gears strategy to realize varying-density multi-scale oil-water micro-drops and emulsions separation. The separation mechanism is proposed based on the different super-spreading behaviors of oil and water, which guarantees a long-time, rapid, continuous, and efficient separation without flux decline. However, it should be noted that there is still room for improvement in the separation efficiency of surfactant emulsions with our dynamic dual-bionic superwetting gears strategy. This is because the high stability of the emulsion makes it difficult to achieve efficient demulsification by the gear squeezed. It may be a promising solution for demulsification by introducing external fields, such as heat, light, or electromagnetic fields. Further efforts from engineering research are needed to expand the size of our device to achieve greater flow, for example, tons per minute. For the 3D printing method, we expect an upgrade in high-precision, large-scale 3D printing technology at high rates and low costs. The dynamic dual-bionic superwetting gears strategy might enable oil-water separation, which would be significant in the renewable energy industry, garbage classification, sewage treatment, and food security.

## Methods
### Materials
The photopolymerized resin, High Temp V2, used for 3D printing, was purchased from Formlabs, United States. Ethanol, tetra-chloromethane, *n*-hexane, and *n*-hexadecane were purchased from Alfa Aesar, United Kingdom. Silicone oil-20 and silicone oil-100 are Sigma-Aldrich. Polydimethylsiloxane (PDMS) was purchased from Dow Corning, Sylgard 184. All chemicals for the chemical resistance test were purchased from Sinopharm, China. Water was acquired from Milli-Q with a resistance of 18.2 MΩ. The biomimetic cat tongue plane with teeth height of 1 mm, pitch of 1 mm, length of 1.15 mm, and tilting angle of 60° were fabricated via bottom-up continuous digital light processing (DLP) using a self-made device powered by a Digital Micromirror Device (DLP Pro6500s, Texas Instruments, USA) at X-Y axes resolution of 33 μm and Z-axis resolution of 10 μm. Printed samples were immersed in ethanol for 10 min and then washed in a 1:1 vol/vol solution of methanol and water. The post-curing process is performed in a tank with 20 W multi-directional LEDs emitting 405 nm light for 5 min at room temperature to enhance mechanical properties. The substrate was treated with $O_2$ plasma (DT-03, OPS Plasma Technology, China) at 200 W for 10 min. Then, superhydrophilic $TiO_2$ nanoparticles (Sigma-Aldrich) dispensed solution as superhydrophilic coating was dip-coated on the surface of printed sample controlled by a motorized vertical mobile device (Mark-10, ESM 301, USA). Super-hydrophilic $TiO_2$ nanoparticles dispensed solution will quickly spread to form the liquid film. Setting the coated printed sample for 10 min at room temperature to evaporate the solvent. And this process is repeated three times to obtain the superhydrophilic printed samples.

PDMS samples were prepared by replicating the morphology of corresponding 3D-printed molds. PDMS mixture was obtained by prepolymer base agent Sylgard 184 A and thermal curing agent Sylgard 184B, 10:1. The mixture was extensively stirred to ensure a homogeneous state and, then degassed in a vacuum oven for 1 h to remove bubbles. Obtained transparent PDMS mixture was poured onto the 3D-printed molds. After degassing, it was cured at 60 °C in an oven for 8 h. At last, the cooled replicates were peeled and stored for further use. The slippery PDMS gear is prepared by immersing it into the corresponding oil phase before the separation usage. XPS survey spectra are shown in Supplementary Fig. 1g–i. All support objects were printed with polylactic acid via fused deposition modeling (FDM, Pro2 Plus, Raise 3D, China) at a Z-axis resolution of 50 μm.

All kinds of O/W emulsions were prepared by mixing oil (tetra-chloromethane, n-hexadecane, and so on) and water in a volume ratio 1:99 under extensive shaking and stirring for 2 h. Then, the mixture is vigorously crushed using an ultrasonic disrupter (30 min) into a stable opaque emulsion.

### Characterization
The optical images of the oil-water micro-drops separation processes were recorded by a digital camera (D7500, Nikon, Japan). X-ray photoelectron spectroscopy (XPS) characterizations confirmed the resin's elemental composition (ESCALab250Xi, ThermoFisher, USA). FTIR spectra were tested by the Fourier transform infrared spectroscopy (VERTEX 70 v, Bruker, USA). SEM images were obtained using a field-emission scanning electron microscope at 10 kV (SU8010, Hitachi, Japan). The samples were cleaned by water and dried by $N_2$ before sputtering a thin layer of platinum (EM ACE, Leica, Germany) to make them electroconductive for SEM imaging. Computed tomography was taken using Bruker SkyScan1272 High-Resolution Micro-CT. Stereo-microscope images were obtained using ZEISS Discovery V8 Stereo-microscope. OLYMPUS BX53 obtained the optical microscopes. The surface morphology of the dual bionic gear surface was obtained by OLYMPUS DSX1000. Contact angles were measured using a contact angle device (DSA 25 S, KRUSS, Germany) with liquid droplets of 3 μL. Each reported contact angle was an average of at least five independent measurements. The liquid adhesion forces were measured by a dynamic contact angle machine (DCAT 20, DataPhysics) with a volume of 3.0 μL for water and oil droplets. The emulsion size was analyzed by the laser granularity instrument (Winner319C, Jinan Winner Particle Instruments Stock Co., Ltd., China).

### Separation mechanism
Two most critical factors in achieving oil-water micro-drop separation are surface wettability and *R* of samples. Theoretically, the competition between oil and water in occupying the surface of PCG is determined

by the difference between $f(\gamma_w \cos\theta_w - \gamma_o \cos\theta_{wo})$ and $\gamma_{ow}$[13,26,47]. Here, $f$ is the surface roughness and equals to $\frac{actual\ ares}{projected\ ares}$. $\gamma_w$, $\gamma_o$ and $\gamma_{ow}$ are surface energies of water-air, oil-air, and water-oil interfaces. For the PCG surface, $n$-hexadecane (27.2 mN m$^{-1}$) can wet it preferentially, while the water phase (72.2 mN m$^{-1}$) is repelled and only spreads on CTG.

Oil and water can form an asymmetric liquid column between two gears. Because of the effect of the surface energy of liquids, two adverse meniscus liquid levels appear at the top and bottom of the liquid column. The liquid will suffer from two additional *Laplace* pressure $P$, $P = \frac{\gamma}{R}\cos\theta$, caused by meniscus liquid level where $\gamma$ and $R$ are the surface tension and RoC of meniscus liquid level, and $\theta$ is the contact angle of the liquid on the flat solid surface[20,43,48,49]. The existence of meniscus liquid levels makes two heterodromous additional *Laplace* pressure $P$. Simultaneously, in consideration of gravity the resultant force along the $Z$-axis, $F_z$ of the liquid column can be determined by Eq. (1):

$$F_z = (P_{up} - P_{down})S - \rho Vg = \gamma\cos\theta\left(\frac{1}{R_{up}} - \frac{1}{R_{down}}\right)S - \rho Vg \quad (1)$$

where $R_{up}$ and $R_{down}$ are RoC of the upper and downward levels, $S$ is the cross-section area of the complementary part in between two gears, $\rho$ and $V$ are the density and volume of the liquid column, and $g$ is the acceleration of gravity. As shown in Fig. 2, the volumes of water and oil are the same, which is equal to $\frac{1}{2}V$. WCA ($\theta_1$) of CTG and OCA ($\theta_2$) of PCG are ~0°. As a result, $F_z$ is obtained:

$$F_z = \frac{1}{2}\left[\left(\frac{\gamma_{w+}\gamma_o}{R_{up}} - \frac{\gamma_{w+}\gamma_o}{R_{down}}\right)S - Vg(\rho_w + \rho_o)\right] \quad (2)$$

where $\gamma_w$ and $\rho_w$ are the surface tension and density of water; $\gamma_o$ and $\rho_o$ are oil surface tension and density of oil. With oil-water micro drops constantly deposited, ever-increasing $V$ makes the direction of $F_z$ switch to the $-Z$-axis resulting in the downflow of the liquid column. $R_{up}$ and $R_{down}$ start getting larger simultaneously, but the change rate of $R_{down}$ is much faster. In the meantime, water and oil spread along two arcs forming two independent liquid films[13]. Finally, the oil-water bridge breaks with a sudden force loss and aggregates into separate water and oil drops. According to the geometrical relationship of models, a smaller $R$ of the arc leads to a more significant difference between $R_{up}$ and $R_{down}$, causing a greater $F_z$. The result is that the liquid downflow speed is too fast to spread along arcs. That's why a more significant $R$ is more beneficial to oil-water separation, even compared with other research of directional liquid transport or oil-water separation[51–53].

## Data availability
All relevant data supporting the key findings of this study are available within the paper and its Supplementary Information or from the corresponding author upon request. Source data are provided with this paper.

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

## Acknowledgements

This work was supported by the National Natural Science Foundation (22122508, 52173293), the National Key Research and Development Program of China (2021YFA070008), Young Elite Scientists Sponsorship Program by China Association for Science and Technology, and the Key Research Program of the Chinese Academy of Sciences (KJZD-EW-M01). We thank F. Chen for the design of the motion stage setup; and J. Peng for the contact angle measurement.

## Author contributions

Z.D. designed the experiments. Z.L., C.L., Y.S. carried out the experiments and performed the data and analysis with the help of Z.Z., T.S., N.L., C.Z., C.Y., L.J. Y.S. and Z.D. wrote the original paper, and Z.D., Z.L. revised it. All authors discussed the results and paper.

## Competing interests

The authors declare no competing interests.
