## [Peer Review File · Nature Communications]

Dual-bionic superwetting gears with liquid directional steering for oil-water separationREVIEWER COMMENTS

Reviewer #1 (Remarks to the Author):

Report on **Co-Designing of Dual-Bionic Superwetting Gears with Liquid Directional Steering for More Sustainable Multi-Scaled Oil-Water Separation**

by Zhuoxing Liu et al.,

Submitted to *Nature Communications*

Summary

Liu et al. developed an oil-water separation device by engaging the two gears with different superwetting features. The oil-water mixture is separated into continuous liquid phases when the mixture droplet is cast on the rotating pre-wetted gears. Authors claim that the critical factors of oil-water separation by this device are (1) the use of two different bionic superwetting surfaces, (2) combining opposite superwetting surfaces in one device, and (3) mechanical separation by gear rotation. This work addressed the flux degradation issues in oil-water separation technology.

Recommendation

This work can reach the standards published in *Nature communications* after major revision.

There is research in applying the bionic directional spreading surfaces for oil-water separation (e.g., *Angew. Chem. Int. Ed.* 2017, 56, 13623.; *Science* 2021, 373, 1344.) or developing the separation devices combining opposite superwetting surfaces (*ACS Appl. Mater. Interfaces* 2015, 7, 18915.). However, the idea of using gears to separate emulsions mechanically is new. The study of the liquid curvature effect and mechanism of microdroplet squeezing is interesting. The manuscript quality is excellent. However, several points should be addressed before publication that I listed thereafter.

Comments for the authors

1. The idea of selling the results under the heading of the dual-bionic surface is intriguing. However, I wonder whether the bionic features are crucial to the oil-water separation. The bionic feature of the nepenthes peristome is directional liquid spreading, while the critical mechanism for this work is lubrication. The surface lubrication of gears originated from the superhydrophilic TiO₂ nanostructure, and oleophilic swelled PDMS, not from the bionic topology. Strictly speaking, the gear surface structure is not the nepenthes peristome but a simple inclined pillar surface. Moreover, the authors lack the demonstration of directional oil spreading onPCT in Figure S3.
2. In Figure 4E, it is unclear how the author decides and compares the fabrication time and material strength. How do authors define the fabrication time and strength of the device? For example, one gear surface is composed of PDMS, whose Young modulus is 0.3-0.8MPa even in unswelled. Device fabrication time is difficult to compare with reference works.
3. What is the effect of PDMS gear swelling on the emulsion separation efficiency? The swelling ratio increases with the oil invasion, which may change the interfacial state (e.g., increased friction,

microdroplet squeezing, etc.).

4. Did the gear nanostructure wear down by the separation process?
5. In Table S2, the authors state that the device exhibits anti-fouling performance. I agree that the device sustainably separates oil and water. However, the anti-fouling performance is not exhibited. I think solid contaminants dissolved in water or oil phase pollute the device.
6. If the hydrophilic gear is first wetted by oil, is it replaced by water?
7. The authors demonstrated the separation of low oil content mixture throughout the study. Is it possible to separate the high-oil content mixture? In addition, is it possible to separate water-in-oil emulsion or that containing surfactant?
8. In Figure 4B, how long does it take to separate the oil-water mixture?
9. In the discussion section, the authors should explain the limitation of this strategy and perspective.

Reviewer #2 (Remarks to the Author):

Comments to the Author

This manuscript reports the dual-bionic super-wetting gears with liquid steering abilities to separate oil-water micro-drops and oil-in-water emulsions into oil and water, respectively. The authors fabricated the fancy automated system with gears that have different topological structures, facilitating the liquid transfer. In particular, the co-designing of the cat tongue plane (CTP) and peristome-inspired cavity plane (PCP) has a positive effect on separation efficiencies, flux, and applicability of various oils with different density because the structures of CTP and PCP fit perfectly to separate oil and water with minimum separation error. To understand the basic principle of oil-water separation, the authors carried out the pre-experiments using pads with the same structure. Based on the variation of the setup construction including pad position, R , and α , the authors also found the optimized condition that can separate the oil/water solution into two solutions (oil and water). As a result, this is a well-organized manuscript, and this work will be of great interest to the related community for further studies of oil/water separators. Therefore, we recommend the manuscript for publication in Nature Communication if the following minor comments can be addressed properly.

Comment 1

We wonder if the hydrophilic gears in Fig. 1d and f are coated with TiO₂ NPs. If the TiO₂ coating was processed on the hydrophilic gears, please include the illustration of the TiO₂ coating process in the 'fabrication process' scheme because TiO₂ coating is essential for super-hydrophilicity.

Comment 2

On page 7, lines 153-154, the author argued 'Our motivation is to utilize the synergies of superwettability, centrifugal force and extrusion force synergies \sim '. Please explain and add in detail what are the synergies of superwettability, centrifugal force, and extrusion force in the separation process, respectively.

Comment 3

On page 11, line 225, the sentence starting 'Form macro-imaging process, teeth and cavities \sim ' has a typing error. Please correct the typos.

Comment 4

In Fig. 3h, the authors used hexadecane and tetrachloromethane with different densities for showing the use of various oils. Please specify each density of oils next to the oil name in the graph.

Comment 5

In Fig. 4c, the collected water looks like a dyed state. To avoid confusion about the image result and separation efficiency that did not separate oil and water perfectly, please supplement the residual components of clean water and the collected water using UV-Vis spectroscopy.

Reviewer #3 (Remarks to the Author):

This manuscript aims to achieve oil-water separation in a more sustainable way. This problem becomes much trickier considering the weak antifouling ability of the existing membrane or sponge separation system and the low feasibility of multi-scale oil-water systems. In their manuscript, the authors take inspiration from the surface morphologies of the cat-tongue and pitcher plant to devise an oil-water separation system.

This work is demonstrated in two parts. First, the authors experimentally compare the parameters by the simple model to screen the structural design in multi-scales. Second, the biomimetic gear's peculiar geometry is assumed to be favorable to the long-termed oil-water separation process. The manuscript combines model experiments with theoretical considerations to rationalize the optimality of the gear design. The authors then test their contraption in real life to prove that their oil-water separation system outperforms existing techniques.

The results they provide are rather convincing. I recommend the publication of this manuscript in Nature Communications.

One issue that needs addressing, is providing more informative descriptions in the caption of Figure 4 and the methods part.

Another point to address is the duration of gears in the long-term experiment.

Reviewer #4 (Remarks to the Author):

The manuscript reports the design and development of a new oil-water separation system based on gears systems with different wettability features that allows oil-water separation via films' spreading-formation-breaking mechanism. The design was well illustrated and the performance investigations were thoroughly performed. Overall, the work offers novelty compared to the established literature, and provides significant contribution to the field of oil spill cleanup and wastewater treatment. Below are some comments and recommendations for revisions to help improve the manuscript:

1. Abstract: Line 29, the wetting is performed before the separation process, please revise this statement according to Lines 109-110.

2. In the introduction section, the authors focused on the existing materials/membranes for oil-water separation but failed to mention and discuss the progress made in scalable oil-water separation systems, that have been recently developed/designed, such as functionalized trawling nets (Environ. Sci.: Water Res. Technol. 2018, 4, 40-47), separators-skimmers (Environ. Technol. Innov. 2020, 18, 100598; Chemosphere 2020, 260, 127586), floating wells/collectors (Langmuir 2021, 37, 6257-6267; N Appl. Sci. 2021, 3, 135), and other devices (Process Saf. Environ. Prot. 2021, 156, 617-624; ACS Appl. Mater. Interfaces 2018, 10, 7504-7511), to name a few. Please revise accordingly to give a better context of the present work in view of the existing literature regarding similar devices and technologies.

3. It is important that the authors clearly state their previous work that this device is partially built on, including "Li, C.; Wu, L.; Yu, C.; Dong, Z.; Jiang, L. Peristome-Mimetic Curved Surface for Spontaneous and Directional Separation of Micro Water-in-Oil Drops. Angew. Chem. Int. Ed. 2017, 56, 13623-13628"

4. Preferentially, the characterization of the prepared CTP and PCP surface (e.g., wettability features) should be discussed before the oil-water separation application for more coherency and to allow readers better establish structure-property-performance relationship.

5. Please explain the cause of the superhydrophilicity and underwater superoleophobicity of CTP. The same for the superoleophilicity and underwater superhydrophobicity of PCP.

6. The chemical characterization (FTIR, XPS, or NMR, ...) of the upper layers/surface of the two gears (CTP and PCP) needs to be performed to investigate the surface chemistry that is vital for the wetting behavior. Please revise accordingly.

7. Movie 1 does not show the entire separation process. It shows the before and after, and the magnified gears connection. The authors need to provide a large-frame and full (sped up) video showing the oil-water separation process (for 1h: ~99 mL of clean water and ~1 mL of red-dyed n-hexadecane, as stated in Lines 119-120).

8. How were the oil-water emulsions prepared before the separation experiments? What is the average size of the oil droplets?

9. The work did not address the simultaneous separation of co-existing heavy and light oil, as

pointed out in Lines 320-342. Please provide additional data (using known mixtures of different oil types in water emulsions) or revise this point about the device performance.

10. The recycling stability of the gears should also be demonstrated by the stability of the gears materials and coatings after a long operating time, such as checking the wettability both gears, and the possibility of TiO₂ nanoparticles leaching that might cause secondary pollution.

11. The effect of temperature on the device's performance is worth investigating to show the impact of oil viscosity change and heating on the proposed films' spread-formation-separation/breaking mechanism. Please revise accordingly.

12. Also, the effect of the gears' rotation speed on the separation efficiency is a very important parameter to investigate to show the effect of residence time needed for efficient films' formation-breaking. Please revise accordingly.

13. What is the composition of the simulated kitchen wastewater?

14. Please add the type of emulsions separated in previous works as an additional column in Extended Data Table 2 to allow a more accurate and relevant comparison.

15. How was the effective surface area calculated for the flux measurements?

16. How can this device be used in practical oil spill cleanup operation in open waters?

17. Please add the missing scale bars in Extended Data Fig. 5.

18. The authors can use a simpler and more straightforward language through the text to better convey their message to the readers. Example (Title): "co-design" or "design"? what is "multi-scaled" referring to? Please revise.

Point-by-point response

Manuscript:

Designing of Dual-Bionic Superwetting Gears with Liquid Directional Steering for More Sustainable Multi-Scaled Oil-Water Separation

The PDF file includes:

Responses to Referee # 1	2
Responses to Referee # 2	11
Responses to Referee # 3	14
Responses to Referee # 4	17

Reviewer comments are set on grey background, the responses to the comments are shown in blue and newly added text in the revised manuscript and SI are shown in green.

Responses to Referee # 1

Referees' Comments:

Summary

Liu et al. developed an oil-water separation device by engaging the two gears with different superwetting features. The oil-water mixture is separated into continuous liquid phases when the mixture droplet is cast on the rotating pre-wetted gears. Authors claim that the critical factors of oil-water separation by this device are (1) the use of two different bionic superwetting surfaces, (2) combining opposite superwetting surfaces in one device, and (3) mechanical separation by gear rotation. This work addressed the flux degradation issues in oil-water separation technology.

Recommendation

This work can reach the standards published in Nature communications after major revision.

There is research in applying the bionic directional spreading surfaces for oil-water separation (e.g., Angew. Chem. Int. Ed. 2017, 56, 13623.; Science 2021, 373, 1344.) or developing the separation devices combining opposite superwetting surfaces (ACS Appl. Mater. Interfaces 2015, 7, 18915.). However, the idea of using gears to separate emulsions mechanically is new. The study of the liquid curvature effect and mechanism of microdroplet squeezing is interesting. The manuscript quality is excellent. However, several points should be addressed before publication that I listed thereafter.

Response: We thank the referee for the positive comments and insightful suggestions to improve the quality of our manuscript. We have updated the manuscript and included a point-to-point response to each individual comment as below.

Comments for the authors

1. *The idea of selling the results under the heading of the dual-bionic surface is intriguing. However, I wonder whether the bionic features are crucial to the oil-water separation. The bionic feature of the nepenthes peristome is directional liquid spreading, while the critical mechanism for this work is lubrication. The surface lubrication of gears originated from the superhydrophilic TiO₂ nanostructure, and oleophilic swelled PDMS, not from the bionic topology. Strictly speaking, the gear surface structure is not the*

nepenthes peristome but a simple inclined pillar surface. Moreover, the authors lack the demonstration of directional oil spreading on PCT in Figure S3.

Response: We greatly appreciate the referee for the valuable suggestions. Topology bionic features are useful and necessary for oil-water separation.

Firstly, complementary topology bionic features can provide rapid transport paths for oil and water separately while ensuring the liquid is squeezed. If there are no topology bionic features, the contact surface of two gears with flat surfaces will be completely closed, greatly reducing the liquid flux, and resulting in liquid spillover. To support our demonstration, we also performed the control experiment with two gears without bionic features. Even if the two gears have the same superwettabilities as the bionic gears, they will still lose the ability to separate oil and water (Fig. R1).

Figure R1(Extended Data Fig. 5). Time sequence images of two gears without topology bionic features for the oil–water micro-drop separation process.

Secondly, the directional transport performance of the topological oil-water paths speeds up the transport rates of oil and water, respectively, thus improving the refresh speed of the two liquid surfaces on gears. This not only increases the flow rates and separation rate but also prevents the deposition of pollutants. The revised manuscript demonstrates the directional water and oil spreading on CTP and PCT (Fig. R2).

Figure R2 (Extended Data Fig. 3). Directional transport of CTP and PCP. a, The wetting of CTP and PCP in a water-oil system. **b,** Directional transport of water on periodically arranged. **c,** Directional transport of oil on PCT.

Thirdly, the directional transport performance based on topology bionic features can also ensure that the liquid films on bionic gears are uniform and dispersed. Therefore, topology bionic features are essential.

Super-lubricity is the key to achieving the separation of oil and water, but it is also inspired by the *Nepenthes* peristome. In conclusion, both superwettability and topology bionic features are indispensable for successful and rapid sustainable multi-scaled oil-water separation.

Our modification to the manuscript: On page 9, lines 210-213, we explicitly described the role of bionic structures in controlling oil-water separation and added references: “Topology bionic features are useful and necessary for oil-water separation, we also performed the control experiment with two gears without bionic features. Even if the two gears have the same superwettabilities as the bionic gears, they will still lose the ability to separate oil and water (Extended Data Fig.5).”

On page 7, lines 151-152, we added described the directional oil spreading on PCT: “and continuous deposition of oil can achieve transport along the arrayed cavity structures directionally (Extended Data Fig. 3).”

2. In Figure 4E, it is unclear how the author decides and compares the fabrication time and material strength. How do authors define the fabrication time and strength of the device? For example, one gear surface is composed of PDMS, whose Young modulus is 0.3-0.8MPa even in unswelled. Device fabrication time is difficult to compare with reference works.

Response: Thanks a lot for the referee’s comments and questions. The preparation time and strength of materials determine the cost and service life of the equipment to some extent. This has a strong reference value for the practical application of oil-water

separation materials. Considering researchers generally did not identify and compare these important parameters in previous relevant studies, we carefully analyzed the experimental section of each literature we compared, calculated the preparation time of the main material, and finally got the data. As for the strength of materials, we also consult and confirm the data after determining the main materials in the literature. So, we believe the comparison results are credible and have strong guiding significance for researchers. Of course, as the reviewer said, the main materials in our experiment included 3D printing resin (hard one) and PDMS (soft one), and the preparation time of the two gears was also different, so we modified the data of this work in Extended Data Fig. 8.

This valuable suggestion greatly improves our manuscript. Thanks again to the reviewers.

Our modification to the manuscript:

Figure R3 (Extended Data Fig. 8) Plots of flux degradation vs. material strength for comparing our method and other separation systems.

3. What is the effect of PDMS gear swelling on the emulsion separation efficiency? The swelling ratio increases with the oil invasion, which may change the interfacial state (e.g., increased friction, microdroplet squeezing, etc.).

4. Did the gear nanostructure wear down by the separation process?

Response: Thanks a lot for the referee’s comments. Comments #3 and #4 are very important questions, we want to reply to these questions together with newly added experiments and characterizations.

The swelling of PDMS gear has a positive effect on improving the separation efficiency and reducing the wear to the nanostructures of the resin gear.

Firstly, PDMS gear swelling can stabilize the oil phase on the gear surface, forming a super lubricating and hydrophobic interface. This ensures that PDMS gear is protected from water for long periods of time, thus achieving sustainable multi-scaled oil-water separation.

Secondly, PDMS gear swelling can reduce the gap between the two gears so that it can better bite, increasing the emulsion's extrusion efficiency. The swollen PDMS gear has good flexibility (Fig. 3d). In the long-term experiment, the flexible PDMS gear can reduce the wear to the rigid resin gear, thus extending the device's service life (Fig. R4).

Figure R4 (Extended Data Fig. 1d, 1f). The microscope images and SEM photographs of the two gears before and after separation. **a-b**, Microscope image of CTG before separation (a) and after separation (b). **c-f**, SEM image of CTG and nano-TiO₂ superhydrophilic coating before separation (c, e) and after separation (d, f). **g-h**, Microscope image of PDMS gear before separation (g) and after separation (h).

In general, PDMS gear swelling is helpful for oil and water separation. The collocation of rigid and flexible complementary materials is also one of the innovation points of our experimental design. We hope that our supplement and explanation can satisfy the reviewers.

Based on the referee's question, a complementary characterization of the durability of gears has been made. After a long separation experiment, we found no obvious reduction of TiO₂ nanoparticles on the surface of the superhydrophilic resin gear.

Our modification to the manuscript: *On page 6, lines 127-129, we added described of gear nanostructure: "Moreover, we found that after a 1h separation experiment, no obvious reduction or damage was observed on the superhydrophilic resin gear or PDMS gear (Extended Data Fig. 1d and 1f)".*

5. In Table S2, the authors state that the device exhibits anti-fouling performance. I agree that the device sustainably separates oil and water. However, the anti-fouling performance is not exhibited. I think solid contaminants dissolved in water or oil phase pollute the device.

Response: Thanks a lot for the referee's comments. Antifouling performance is very important for oil-water separation. Good antifouling performance can inhibit the decline of flow and separation efficiency in the separation process, thus greatly prolonging the service life. In our experiment, our dual-bionic superwetting gears device shows little influence in separation flux even after 600 minutes of separation, demonstrating the anti-fouling features.

Firstly, the whole device is an open system, so it is not as prone to blockage and deposition by pollutants as the oil-water separation membrane or sponge. According to the reviewer's suggestions, solid contaminants are important for proofing our demonstration. So, we supplemented the solid pollutant experiment. We used insoluble solid particles of different sizes to simulate solid pollutants ($r = 0.5$ mm, 1 mm, 1.5 mm, and 2 mm). The solid particles were put into the *n*-hexadecane-water emulsion, and the separation experiment was carried out after intense stirring. Our new separation mode can quickly remove solid pollutants on the gear surfaces with constantly emerging liquid films and continuous moving liquid flow. (**Figure R5**)

Secondly, underwater superoleophobic, underoil superhydrophobic materials have proven to have very good anti-fouling performance^{R1-R4}, which can prevent the adhesion of liquid oil or water phase. Prolonged separation does not result in the dual-bionic superwetting gears device clogging, and the separation flow does not decrease. After separation, there are no significant solid particle pollutants on the surface of the gear or

damage to the surface morphology (**Figure R4**). The result shows that our dynamic open separation model has a good antifouling effect.

Figure R5. The anti-fouling performance of separation device. **a**, Microscope photographs of the solid contaminants. The radii are 0.5mm, 1mm, 1.5mm and 2mm respectively. **b-c**, The time sequence images of the separation process for solid contaminants. **b**, $r = 1$ mm **c**, $r = 2$ mm (**c**) **d**, Separation of solid contaminants of different sizes.

- R1. Li, N., *et al.* Superamphiphilic TiO₂ composite surface for protein antifouling. *Adv. Mater.*, **34**, 2003559 (2021).
- R2. Wang, D., *et al.* Seawater-induced healable underwater superoleophobic antifouling coatings. *ACS Appl. Mater. Interfaces*, **11**, 1353-1362 (2018).
- R3. Song, H., *et al.* Salt-induced and alcohol-induced hydrophobic and underoil superhydrophobic poly (vinylidene fluoride) membranes for effective oil collection. *J. Environ. Chem. Eng.*, **9**, 104714 (2021).
- R4. Ding, D., *et al.* Underwater superoleophobic-underoil superhydrophobic Janus ceramic membrane with its switchable separation in oil/water emulsions. *J. Membr. Sci.*, **565**, 303-310 (2018).

6. If the hydrophilic gear is first wetted by oil, is it replaced by water?

Response: Oil cannot be replaced by water because the swelling oil can protect the gear from wetting by the water phase. In the fabrication process, we pre-wet each gear with water or oil before separating. In the same way, the hydrophilic gear will not be replaced with oil when it is wetted by water. So, our dual-bionic superwetting gears device can achieve long time oil-water separation without any loss of efficiency.

7. The authors demonstrated the separation of low oil content mixture throughout the study. Is it possible to separate the high-oil content mixture? In addition, is it possible to separate water-in-oil emulsion or that containing surfactant?

9. In the discussion section, the authors should explain the limitation of this strategy and perspective.

Response: Thanks for the referee's valuable comment and professional question. Comments #7 and #9 are very meaningful, which can make our paper more objective, and at the same time, can give readers more inspiration, thus encouraging future research.

Response to comment #7: Improving the oil content for the mixture of oil-water microdroplets will not affect the separation efficiency and speed. However, it is a big challenge to separate emulsions containing surfactants. The separation efficiency was not as high as expected, so we did not present these results in detail in this paper. This is a limitation of our study. At the same time, this is a research project that we will focus on in future research. Following the reviewer's suggestion, we rewrote the **Discussion section** to highlight our research's limitations.

Response to comment #9: We rewrote the Discussion section in the revised manuscript according to the referee's suggestion. Moreover, we present some perspectives to guide researchers to make breakthroughs based on our work in the future.

***Our modification to the manuscript:** On page 20, lines 404-418, we rewrote the discussion part: "In summary, we have presented a dynamic dual-bionic superwetting gears strategy to realize varying-density multi-scale oil-water micro-drops and emulsions separation. The separation mechanism is proposed based on the different super-spreading behaviors of oil and water, which guarantees a long-time, rapid, continuous, and efficient separation without flux decline. However, it should be noted that there is still room for improvement in the separation efficiency of surfactant emulsions with our dynamic dual-bionic superwetting gears strategy. This is because the high stability of the emulsion makes it difficult to achieve efficient demulsification by the gear squeezed. It may be a promising solution for demulsification by introducing external fields, such as heat, light, or electromagnetic fields. Further efforts from engineering research are needed to expand the size of our device to achieve greater flow, for example, tons per minute. For the 3D printing method, we expect an upgrade in high-precision, large-scale 3D printing technology at high rates and low costs. The dynamic dual-bionic superwetting gears*

strategy generally opens a brand-new mode of multi-scaled oil-water separation, which will be significant in the renewable energy industry, garbage classification, sewage treatment, and food security.”

8. In Figure 4B, how long does it take to separate the oil-water mixture?

Response: Thanks a lot for the referee’s comments. We have modified this part in our revised manuscript, as described below:

Our modification to the manuscript: *On Page 16, Line 335-336, we added a discussion of the separation time; “The separation rate was set at $40 \mu\text{L s}^{-1}$, and the total separation time was about 420 minutes.”*

Responses to Referee # 2

Referees' Comments:

Comments to the Author

This manuscript reports the dual-bionic super-wetting gears with liquid steering abilities to separate oil-water micro-drops and oil-in-water emulsions into oil and water, respectively. The authors fabricated the fancy automated system with gears that have different topological structures, facilitating the liquid transfer. In particular, the designing of the cat tongue plane (CTP) and peristome-inspired cavity plane (PCP) has a positive effect on separation efficiencies, flux, and applicability of various oils with different density because the structures of CTP and PCP fit perfectly to separate oil and water with minimum separation error. To understand the basic principle of oil-water separation, the authors carried out the pre-experiments using pads with the same structure. Based on the variation of the setup construction including pad position, R , and α , the authors also found the optimized condition that can separate the oil/water solution into two solutions (oil and water). As a result, this is a well-organized manuscript, and this work will be of great interest to the related community for further studies of oil/water separators. Therefore, we recommend the manuscript for publication in Nature Communication if the following minor comments can be addressed properly.

Response: We thank the referee for these positive comments. We have updated the manuscript and included a point-to-point response to each individual comment as below.

1. We wonder if the hydrophilic gears in Fig. 1d and f are coated with TiO_2 NPs. If the TiO_2 coating was processed on the hydrophilic gears, please include the illustration of the TiO_2 coating process in the 'fabrication process' scheme because TiO_2 coating is essential for super-hydrophilicity.

Response: Thanks a lot for the referee's comments and suggestions. Based on the referee's suggestion, we have expanded the relevant content of the **Methods section**.

Our modification to the manuscript: Under the Method, on Page 21, Line 434-440, we explicitly described the fabrication process of the superhydrophilic TiO_2 nanoparticles coating on the gear structures: "Then, superhydrophilic TiO_2 nanoparticles dispensed solution (Sigma-Aldrich) as superhydrophilic coating was dip-coated on the surface of

printed sample controlled by a motorized vertical mobile device (Mark-10, ESM 301, USA). Superhydrophilic TiO₂ nanoparticles dispensed solution will quickly spread to form the liquid film. Setting the coated printed sample for 10 minutes at room temperature to evaporate the solvent. And this process is repeated three times to obtain the superhydrophilic printed samples.”

2. On page 7, lines 153-154, the author argued ‘Our motivation is to utilize the synergies of superwettability, centrifugal force and extrusion force synergies ~’. Please explain and add in detail what are the synergies of superwettability, centrifugal force, and extrusion force in the separation process, respectively.

Response: Thanks a lot for the referee’s comments and suggestions. The synergies of superwettability, centrifugal force, and extrusion force in the separation process are discussed in detail in the **Separator Design and Working Mechanism Section** (Page 11-13). Moreover, Figure 3 illustrates these three synergies' separation mechanisms and their roles in the separation process. To make the paper have clearer expression and more concise logic, we did not give too much description on this point in the front part of our paper. The referee’s suggestion of a short discussion would help readers better understand the experiment. Based on the referee’s suggestion, we explain the synergies of superwettability, centrifugal force, and extrusion force (page 8, lines 163-166) concisely.

Our modification to the manuscript: *Based on the referee’s suggestion, we modified this part in the revised manuscript on page 8, lines 163-166: “Our motivation is to utilize the synergies of superwettability (to realize the overspread of different liquids on different surfaces and guide the liquids’ directional transport), centrifugal force (to accelerate liquids flow, promote the liquid bridge breaking) and extrusion force (to break the dispersed phase of the emulsion) synergies to enable rapid, continuous, and efficient oil-water mixture and oil-in-water emulsion separation.”*

3. On page 11, line 225, the sentence starting ‘Form macro-imaging process, teeth and cavities ~’ has a typing error. Please correct the typos.

Response: We have replaced “Form” by “From” on page 11, line 240 in the revised manuscript.

4. In Fig. 3h, the authors used hexadecane and tetrachloromethane with different densities for showing the use of various oils. Please specify each density of oils next to the oil name in the graph.

Response: Thanks for the referee's suggestions. In the revised manuscript, we added the densities of hexadecane and tetrachloromethane in Figure 3h.

Figure R6 (Figure 3h). h, Plots of flux degradation for separating hexadecane-water and tetrachloromethane systems.

5. In Fig. 4c, the collected water looks like a dyed state. To avoid confusion about the image result and separation efficiency that did not separate oil and water perfectly, please supplement the residual components of clean water and the collected water using UV-Vis spectroscopy.

Response: Thanks a lot for the referee's comments and suggestions. Based on the referee's suggestion, we have supplemented the characterization analysis using UV-Vis spectroscopy based on the referee's suggestion. To accurately reflect the relationship between *n*-Hexadecane concentration and absorbance (Fig. R7a), we got the standard curve: $A = 0.00659 C + 0.03743$ ($R^2 = 0.99876$). And we test the absorbance of the collected water and the purified water, the result showed that the extremely low oil content in the collected water is 0.052 mg/L. We hope that our results can satisfy the reviewers.

Figure R7. UV-Vis spectra of the collected water and the purified water. a, Standard curves for different concentrations of *n*-Hexadecane. b, The collected water after the separation process matches the purified water, indicating high separation efficiency.

Responses to Referee # 3

Referees' Comments:

This manuscript aims to achieve oil-water separation in a more sustainable way. This problem becomes much trickier considering the weak antifouling ability of the existing membrane or sponge separation system and the low feasibility of multi-scale oil-water systems. In their manuscript, the authors take inspiration from the surface morphologies of the cat-tongue and pitcher plant to devise an oil-water separation system.

This work is demonstrated in two parts. First, the authors experimentally compare the parameters by the simple model to screen the structural design in multi-scales. Second, the biomimetic gear's peculiar geometry is assumed to be favorable to the long-termed oil-water separation process. The manuscript combines model experiments with theoretical considerations to rationalize the optimality of the gear design. The authors then test their contraption in real life to prove that their oil-water separation system outperforms existing techniques.

The results they provide are rather convincing. I recommend the publication of this manuscript in Nature Communications.

Response: We thank the referee for the positive comments and insightful suggestions to improve the quality of our manuscript. We have updated the manuscript and included a point-to-point response to each individual comment as below.

1. One issue that needs addressing, is providing more informative descriptions in the caption of Figure 4 and the methods part.

Response: Based on the referee's suggestion, we have expanded the caption of Figure 4, hoping to provide readers with comprehensive and useful information.

Our modification to the manuscript:

“Figure 4. Dual-bionic gears for sustainable oil-water separation. a, Schematic of the integrated dual-bionic gears separator for oil-in-water (O/W) emulsion. The pump sucks up O/W emulsion to the entrance of the separator. Through the extrusion and centrifugation process with the assistance of a brush, pure oil or water phase can be collected from two exits. b, Top: Experimental setup of the dual-bionic gears for sustainable oil-water separation. Bottom: Optical and stereomicroscopic images of O/W emulsion and collected water and oil. A histogram picture of emulsion particle size distribution. In a magnified stereomicroscope image, the opaque emulsion is separated

into clear water and clear oil phases after the separation. **c**, Separation efficiencies of various oil-water mixtures with different densities, viscosity, and surface tension. **d**, The proof-of-concept experiment of dual-bionic superwetting gears device to separate water and oil in a swimming pool. **e**, Separation efficiencies of different oil-water micro-drops. Inset, the collection of simulated kitchen wastewater. There are three types of simulated kitchen wastewater: 1 M NaCl/salad oil, H₂O/chili oil, and soy sauce/salad oil.”

In addition, we have also revised the **Methods** part in the revised manuscript on Page 21, lines 451-454, as below: “All kinds of O/W emulsions were prepared by mixing oil (tetrachloromethane, n-hexadecane, and so on) and water in a volume ratio 1:99 under extensive shaking and stirring for 2 h. Then, the mixture is vigorously crushed using an ultrasonic disrupter (30 min) into a stable opaque emulsion.”

2. Another point to address is the duration of gears in the long-term experiment.

Response: Thanks for the referee’s suggestions. Based on the referee’s suggestion, a complementary characterization of the durability of gears has been made. After a long separation experiment, we found no obvious reduction of TiO₂ nanoparticles on the surface of the superhydrophilic resin gear. We also characterized the surface morphology of PDMS gear before and after a long time separation experiment, and found that there was little change, as shown in Figure R4:

Figure R4 (Extended Data Fig. 1d, 1f). The microscope images and SEM photographs of two gears before and after separation. a-b, Microscope image of CTG before separation (a) and after separation (b). **c-f**, SEM image of CTG and nano-TiO₂

superhydrophilic coating before separation (c, e) and after separation (d, f). **g-h**, Microscope image of PDMS gear before separation (g) and after separation (h).

Figure R8. Underwater hexadecane contact angle on the CTG (Left) and under-hexadecane water contact angle on the PCG (Right).

At the same time, we also characterized the changes in the wettability of the two gears after a long-time separation. The results of the contact angle test proved that the wettability of the two gears did not change.

In summary, our dual-bionic superwetting gears device has good durability and can be used for long time separation experiments.

Responses to Referee # 4

Referees' Comments:

The manuscript reports the design and development of a new oil-water separation system based on gears systems with different wettability features that allows oil-water separation via films' spreading-formation-breaking mechanism. The design was well illustrated and the performance investigations were thoroughly performed. Overall, the work offers novelty compared to the established literature, and provides significant contribution to the field of oil spill cleanup and wastewater treatment. Below are some comments and recommendations for revisions to help improve the manuscript:

Response: We thank the referee for insightful suggestions to improve the quality of our manuscript. We have updated the manuscript and included a point-to-point response to each individual comment as below.

1. Abstract: Line 29, the wetting is performed before the separation process, please revise this statement according to Lines 109-110.

Response: Thanks for the referee's comments and suggestions. The pure water or oil phase is pre-wet on the gear surface before the separation process. Then, during the separation process, the water or oil phase in the mixture can steer to spread rapidly and continuously on the water or oil pre-wetted gear surface.

Our modification to the manuscript: On page 2, line 29, "the water and oil in the mixture can steer to spread on the preferential gear surfaces rapidly and continuously."

2. In the introduction section, the authors focused on the existing materials/membranes for oil-water separation but failed to mention and discuss the progress made in scalable oil-water separation systems, that have been recently developed/designed, such as functionalized trawling nets (Environ. Sci.: Water Res. Technol. 2018, 4, 40-47), separators-skimmers (Environ. Technol. Innov. 2020, 18, 100598; Chemosphere 2020, 260, 127586), floating wells/collectors (Langmuir 2021, 37, 6257-6267; N Appl. Sci. 2021, 3, 135), and other devices (Process Saf. Environ. Prot. 2021, 156, 617-624; ACS Appl. Mater. Interfaces 2018, 10, 7504-7511), to name a few. Please revise accordingly to give a better context of the present work in view of the existing literature regarding similar devices and technologies.

Response: Thanks for the referee’s professional comments and suggestions. The referee’s suggestion can enhance the discussion of our introduction part with scalable oil-water separation systems. So, we modified the **Introduction** and added new **References** in the revised manuscript.

Our modification to the manuscript: On page 3, lines 57-62, “*Scalable oil-water separation devices have progressed in recent years, including functionalized trawling nets¹⁹, large separators-skimmers²⁰, floating wells²¹, etc. But the underlying mechanism of these devices is also based on superwetting membranes or sponges.²⁰⁻²³ Although the total amount or separation rate has been improved, the limitations of the closed and static separation modes are not broken. The above three deficiencies still exist.*”

19. Barry, E. et al. Mitigating oil spills in the water column. *Environ. Sci.: Water Res. Technol.* **4**, 40-47 (2018).
20. Abidli, A., Huang, Y., Cherukupally, P., Bilton, A. M., & Park, C. B. Novel separator skimmer for oil spill cleanup and oily wastewater treatment: From conceptual system design to the first pilot-scale prototype development. *Environ. Technol. Innov* **18**, 100598 (2020).
21. Yan, X., Liu, G., Xu, J., & Ma, X. In situ oil separation and collection from water under surface wave condition. *Langmuir* **37**, 6257-6267 (2021).
22. Abidli, A., Huang, Y., & Park, C. B. In situ oils/organic solvents cleanup and recovery using advanced oil-water separation system. *Chemosphere* **260**, 127586 (2020).
23. Choe, Y., et al. Gravity-based oil spill remediation using reduced graphene oxide/LDPE sheet for both light and heavy oils. *Process Saf. Environ. Prot.* **156**, 617-624 (2021).

We hope that our modification can satisfy the reviewers.

3. It is important that the authors clearly state their previous work that this device is partially built on, including “Li, C.; Wu, L.; Yu, C.; Dong, Z.; Jiang, L. Peristome-Mimetic Curved Surface for Spontaneous and Directional Separation of Micro Water-in-Oil Drops. *Angew. Chem. Int. Ed.* 2017, **56**, 13623-13628”.

Response: Thanks for the referee’s suggestion. This is a good reminder for us to introduce our previous paper. We added a short discussion in the revised manuscript based on the referee's comment.

Our modification to the manuscript: On Page 5, lines 110-111, “*Based on the static dual-curved oil-water microdroplet separation device we reported earlier,¹³ the upgraded dynamic dual-bionic superwetting gears device has been successfully designed.*”

4. Preferentially, the characterization of the prepared CTP and PCP surface (e.g., wettability features) should be discussed before the oil-water separation application for more coherency and to allow readers better establish structure-property-performance relationship.

Response: Thanks for the referee's comments and suggestions. The reviewer's suggestions are very professional and worthy of our serious consideration. As to the characterization of the prepared CTP and PCP surface, we have shown the results in **Extended Data Fig. 1**.

Different from referee's comments, we did not display and analyze these characterizations in the **Designed dual-bionic gears for oil-water separation** section, mainly for two reasons:

Firstly, we want to be able to introduce our design concept to readers in the most concise and clear sentences, from the plane to the curve to the gear. So, we don't present a concrete characterization here, just a result. Detailed characterization is, therefore, shown in the **Screening dual-bionic gears designed for sustained oil-water separation** section;

Secondly, in **Screening dual-bionic gears designed for sustained oil-water separation section**, we present detailed experiments and discuss the effects of various parameters. It would be easier for the reader to understand. Just as the reviewer mentioned the "structure-property-performance relationship", that's how our part of the logic works.

Thanks again for the referee's advice. We hope that our supplement can satisfy the referee.

5. Please explain the cause of the superhydrophilicity and underwater superoleophobicity of CTP. The same for the superoleophilicity and underwater superhydrophobicity of PCP.

Response: Thanks for the referee's professional comments. This is a very professional question, and the supplement and explanation in this respect can make the basic theory of our paper more clearly. Therefore, we add relevant explanations and a schematic diagram (Figure R10). For the superhydrophilic CTP, as we wrote in the **Separation mechanism** section, which was determined by the difference between $f(\gamma_w \cos \theta_w - \gamma_o \cos \theta_o)$ and γ_{ow} . The $\theta_{o'}$ can be obtained by formula (eq R1)⁶¹,

$$\cos \theta_{o'} = \frac{f(\gamma_w \cos \theta_w - \gamma_o \cos \theta_o)}{\gamma_{ow}} \quad (\text{eq R1})$$

Where $\theta_o = \theta_w = 0^\circ$, $\gamma_o < \gamma_w$, and considering the $f < 1$. We can calculate that $\theta_{o'} > 150^\circ$. Therefore, CTP exhibits excellent underwater superoleophobicity.

For the PCP surface, similar results can be obtained by (eq R2). Here $\theta_o = 0^\circ$. Therefore, $\theta_{w'}$ will be decided by θ_w and f . And we know that θ_w of PDMS is greater than 90° .

$$\cos \theta_{w'} = \frac{f(\gamma_o \cos \theta_o - \gamma_w \cos \theta_w)}{\gamma_{wo}} \quad (\text{eq R2})$$

On the other hand, in the experiment, the PDMS will first swell with oil. So, its surface is filled with oil, and there is no trapped air. When immiscible water droplets with high surface energy are introduced, the solid surface submerged by oil can support the oil-water interface, greatly reducing the water-solid contact area (f). This means the droplets float rather than invade and spread. Thus, PCP exhibits superhydrophobicity underoil.

Our modification to the manuscript:

Figure R9. The schematic diagrams of the CTP's Underwater Superoleophobicity and PCP's underoil superhydrophobicity.

61. Liu, M., *et al.* Bioinspired design of a superoleophobic and low adhesive water/solid interface. *Adv. Mater.*, **21**, 665-669 (2009).

6. The chemical characterization (FTIR, XPS, or NMR, ...) of the upper layers/surface of the two gears (CTP and PCP) needs to be performed to investigate the surface chemistry that is vital for the wetting behavior. Please revise accordingly.

Response: Thanks for the referee's suggestions. We supplemented the chemical characterization (FTIR and XPS) of the upper layers/surface of the two gears in the revised Supporting Information, as described below:

Figure R10. Chemical composition analysis of 3D printed cat tongue structure and replicating PDMS peristome-inspired cavity structure. a-c, FTIR analysis. a, FTIR spectra for resin (black) and resin + H₂O (red). b, FTIR spectra for PDMS (black), PDMS + *n*-hexadecane (red) and PDMS + chili oil (dark red). c, FTIR spectra for the pdms immersed with different oil phase. d-g, XPS analysis. d, XPS full spectrum of PDMS (purple) and resin (blue). e, Si 2p pattern of PDMS. f, Si 2p pattern of resin. g, Ti 2p pattern of resin.

The FT-IR spectra showed that the resin was successfully wetted by water and oil swelled into PDMS. Wetted resin showed a strong stretching vibration peak at 3305 cm⁻¹ and a

bending vibration peak at 1639 cm^{-1} attributed to $-\text{OH}$. The PDMS with the n-hexadecane invasion showed two stretching vibration peaks near 2925 cm^{-1} and a bending vibration peak at 1466 cm^{-1} attributed to $-\text{CH}_2$. The PDMS with chili oil invasion showed a stretching vibration peak at 1747 cm^{-1} assigned to $\text{C}=\text{O}$ besides the two stretching vibration peaks near 2925 cm^{-1} . The infrared spectrum of PDMS immersed with other oils was shown in Fig. R11c.

The elemental composition of the wetted resin and PDMS are confirmed by X-ray photoelectron spectroscopy (XPS) characterizations. XPS survey spectra showed that several typical peaks for O 1s, Ti 2p, C 1s, and Si 2p can be clearly identified in the resin with a coating of TiO_2 nanoparticles while O 1s, C 1s, Si 2p, and O 2s in PDMS. Figure R10e showed the Si2p characteristic peak of the PDMS. In the XPS spectra of resin, as shown in Fig. R11 f-g, the intensity of peaks at 103.18 eV correspond to the Si-O bonds while 458.63 eV and 464.48 eV correspond to the Ti-O bonds, respectively. It shows that cat tongue mimetic photocurable resin teeth with a coating of TiO_2 nanoparticles.

Our modification to the manuscript: *Based on the referee's suggestion, XPS survey spectra are shown in Extended Data Fig. 1: On Page 27, lines 646-649, "g, XPS full spectrum of PDMS (purple) and resin (blue). h, Si 2p pattern of resin surface. i, Ti 2p pattern of resin surface. XPS survey spectra showed that several typical peaks for O 1s, Ti 2p, in the resin. The intensity of peaks at 103.18 eV corresponds to the Si-O bonds while 458.63 eV and 464.48 eV correspond to the Ti-O bonds, respectively."*

On Page 29, lines 651-659, FTIR spectra are shown in Extended Data Fig. 2 "b, FTIR spectra for resin (black) and resin + H₂O (red). c, FTIR spectra for PDMS (black), PDMS + n-hexadecane (red) and PDMS + chili oil (dark red). The PDMS with the n-hexadecane invasion showed two stretching vibration peaks near 2925 cm^{-1} and a bending vibration peak at 1466 cm^{-1} attributed to $-\text{CH}_2$. The PDMS with chili oil invasion showed a stretching vibration peak at 1747 cm^{-1} assigned to $\text{C}=\text{O}$ besides the two stretching vibration peaks near 2925 cm^{-1} . d, FTIR spectra for the PDMS immersed with different oil phases. The FTIR spectra showed that the resin was successfully wetted by water and wetted resin showed a strong stretching vibration peak at 3305 cm^{-1} and a bending vibration peak at 1639 cm^{-1} attributed to $-\text{OH}$."

*And we added the **Methods** section: On Page 21, lines 457-460, "X-ray photoelectron spectroscopy (XPS) characterizations confirmed the resin's elemental composition (ESCALab250Xi, ThermoFisher, USA). FTIR spectra were tested by the Fourier*

transform infrared spectroscopy (VERTEX 70v, Bruker, USA)."

We hope that our modification can satisfy the reviewers.

7. Movie 1 does not show the entire separation process. It shows the before and after, and the magnified gears connection. The authors need to provide a large-frame and full (speed up) video showing the oil-water separation process (for 1h: ~99 mL of clean water and ~1 mL of red-dyed n-hexadecane, as stated in Lines 119-120).

Response: Thanks for the referee's comments. We have provided a large-frame and full (speed-up 30X) video showing the oil-water separation process in the revised Supporting Information (Movie 2), as described below:

Figure R11. Time sequence images of the oil-water separation process. **a**, 0 min. **b**, 15 min. **c**, 30 min. **d**, 60 min.

Our modification to the manuscript: On page 6, lines 126-127, "After 1h, ~99 mL of clean water and ~1 mL of red-dyed n-hexadecane are collected (Fig. 1g, Movie 2)."

8. How were the oil-water emulsions prepared before the separation experiments? What is the average size of the oil droplets?

Response: Thanks for the referee's suggestions. We have added related content to describe the preparation of oil-water emulsions in the **Methods** section in the revised manuscript on Page 21, lines 451-454. In addition, the emulsion dispersing phase's particle size and distribution (Figure R12) were analyzed by the laser granularity instrument (Winner319C) and the average size of the oil droplets is 100 μm .

Our modification to the manuscript: On Page 21, lines 451-454, "All kinds of O/W emulsions were prepared by mixing oil (tetrachloromethane, n-hexadecane, and so on) and water in a volume ratio 1:99 under extensive shaking and stirring for 2 h. Then, the mixture is vigorously crushed using an ultrasonic disrupter (30 min) into a stable opaque

emulsion.”

And we added the Methods section: On Page 22, lines 471-72, “The emulsion size was analyzed by the laser granularity instrument (Winner319C, Jinan Winner Particle Instruments Stock Co., Ltd., China).”

On page 16, lines 329-331, we added described the particle size of the emulsion: “In addition, the emulsion dispersing phase's particle size and distribution (Fig. 4c) were analyzed by the laser granularity instrument (Winner319C) and the average size of the oil droplets is $\sim 100 \mu\text{m}$.”

Figure R12 The particle size of the emulsion. a,c, Schematic of emulsion before separation (a) and after separation (c). b, Histogram of emulsion particle size distribution. d, e, Optical and stereomicroscopic images of collected water (d) and collected oil (e).

9. The work did not address the simultaneous separation of co-existing heavy and light oil, as pointed out in Lines 320-342. Please provide additional data (using know mixtures of different oil types in water emulsions) or revise this point about the device performance.

Response: Thanks for the referee’s constructive suggestion. We supplemented a synchronous separation experiment via our dual-bionic superwetting gears device to simulate the co-existing heavy and light oil situation. The heavy oil/water emulsion and the light oil/water emulsion dripped into the middle of both gears simultaneously. After a period of separation, pure water and a mixture of the two oils can be obtained, respectively. This experimental result proves that our device can simultaneously separate the mixtures

of different oil types in water emulsions. The whole experiment process and results are added in the revised Supporting Information.

Our modification to the manuscript: On Page 16, lines 344-345, “Our dual-bionic separation device can solve this problem effectively (Extended Data Fig. 7).”

Figure R13 (Extended Data Fig. 7). Optical sequence images of two gears for the oil–water emulsion separation with co-existing of both heavy and light oils, i.e., O-W emulsion 1 system including tetrachloromethane (heavy oil, density of 1.59 g cm^{-3}) in the water phase and O-W emulsion 2 system including hexadecane (light oil, density of 0.77 g cm^{-3}) in the water phase. Both oils are dyed by red to enhance visualization.

10. The recycling stability of the gears should also be demonstrated by the stability of the gears materials and coatings after a long operating time, such as checking the wettability both gears, and the possibility of TiO_2 nanoparticles leaching that might cause secondary pollution.

Response: Thanks for the referee’s comments. This is a very important and meaningful question. A complementary characterization of the durability of gears has been made. After a long separation experiment, we found TiO_2 nanoparticles on the surface of the superhydrophilic resin gear were still there, and no obvious reduction was observed. We also characterized the surface morphology of PDMS gear before and after a long-time separation experiment, and found that there was little change, as described below:

Figure R4 (Extended Data Fig. 5). The microscope images and SEM photographs of two gears before and after separation. **a-b**, Microscope image of CTG before separation (a) and after separation (b). **c-f**, SEM image of CTG and nano-TiO₂ superhydrophilic coating before separation (c, e) and after separation (d, f). **g-h**, Microscope image of PDMS gear before separation (g) and after separation (h).

Figure R8. Underwater hexadecane contact angle on the CTG (Left) and under-hexadecane water contact angle on the PCG (Right).

At the same time, we also characterized the changes in the wettability of the two gears after a long-time separation experiment. The results of the contact angle test proved that the wettability of the two gears did not change.

To sum up, our dual-bionic superwetting gears device has good durability and can be used

for long time separation experiments.

11. The effect of temperature on the device’s performance is worth investigating to show the impact of oil viscosity change and heating on the proposed films’ spread-formation-separation/breaking mechanism. Please revise accordingly.

Response: Thanks for the referee’s comments. First, we demonstrate that the wettability of our dual-bionic superwetting gears can remain stable within a certain temperature range. We placed the two gears at 20°C, 50°C, and 80°C for 2 hours, then measured their contact angles, respectively. It turns out that they still maintain good underwater superoleophobicity, and underoil superhydrophobicity as described below:

Figure R14. Underwater hexadecane contact angle on the CTG (up) and under-hexadecane water contact angle on the PCG (down) at different temperature.

Table R1. The viscosity of oil at different temperature.

Various oil	Viscosity (mPa · s)		
	20°C	50°C	80°C
n -Hexadecane	3.27	2.19	1.37
Tetrachloromethane	1.96	0.74	0.65
n -Hexane	0.33	0.33	—
FC-72	2.49	0.69	—
Silicone oil-20	22.3	13.8	8.81
Silicone oil-100	146.17	81.6	4.58
Salad oil	55.3	25.23	11.3
Chili oil	65.7	36.13	12.5

Then, we tested the viscosity of different oils at 20°C, 50°C and 80°C (Table R1). Next,

silicone oil-100/water emulsion via our dual-bionic superwetting gears at three different temperatures (30 °C, 40 °C, and 55 °C) showed that the temperature rise has no negative effect on the separation efficiency (Figure R15). We think this is because temperature changes have little effect on water viscosity. In addition, in general, the higher the temperature, the lower the viscosity of the oil phase. Therefore, less resistance and demulsification are easier to achieve in the extrusion process. We need to mention that a higher ambient temperature means faster liquid volatilization and more liquid loss. So, the oil and water separation in a high-temperature environment is of little significance for improving separation in our system.

Figure R15. Optical and thermal images of oil-water separation at different temperatures.

12. Also, the effect of the gears' rotation speed on the separation efficiency is a very important parameter to investigate to show the effect of residence time needed for efficient films' formation-breaking. Please revise accordingly.

Response: Thanks for the referee’s comments. We add the test of the influence of rotation speed. We tested the separation efficiency of n-hexadecane/water emulsion at different rotation speeds within our device capacity. The results show that the efficiency of emulsion separation is very low when the rotational speed is very small. This may be because too much of the dispersed oil phase falls down the gear before it can be squeezed. When the rotational speed is increased, the emulsion separation efficiency can be maintained at a very high level. The supplement of this question makes our research more systematic and comprehensive. Thank you very much for your help.

Figure R16 (Extended Data Fig.6). Optical sequence images of two gears for the oil–water emulsion separation process at different rotation speeds.

Our modification to the manuscript: On Page 16, lines 336-339, “Besides, we tested and compared the separation efficiency of n-hexadecane/water emulsion at different rotation speeds within our device capacity. The results showed that the maximum separation efficiency is 99.4% when two gears with opposite rotation directions by an angular velocity ω of 6 rpm (Extended Data Fig. 6).”

13. What is the composition of the simulated kitchen wastewater?

Response: Based on the referee’s suggestion, we have explained the composition of the simulated kitchen wastewater in the revised manuscript on Page 18, lines 380-381; “We simulated three groups of kitchen wastewater micro-drops: 1 M NaCl/salad oil, H₂O/chili oil, and soy sauce/salad oil.”

14. Please add the type of emulsions separated in previous works as an additional column in Extended Data Table 2 to allow a more accurate and relevant comparison.

Response: Thanks for the referee's comments. The type of emulsions has been added to the **Extended Data Table 2** in the revised Supporting Information.

Our modification to the manuscript: On page 38

Extended Data Table 2 | Comparison of oil-water separation ability reported in the recent literatures with our work

Main materials	Wettability	Emulsion separation	Emulsion type	External field	Separation rate	Anti-fouling	Ref.
Hydrogel grafted PVDF	Superhydrophilic/ Underwater superoleophobic	√	O/W	√	25000 L m ⁻² h ⁻¹ bar ⁻¹	√	37
TiO ₂ , TEOS	Superhydrophilic/ Superoleophobic	√	O/W	-	1200 L m ⁻² h ⁻¹	-	50
Superamphiphilic, SiO ₂ -TiO ₂	Superamphiphilic/ Underwater superoleophobic	√	O/W O/O	-	-	-	39
Cu, PDMS	Underwater superamphiphilic	√	O/W	-	1.128 mL cm ⁻² s ⁻¹	-	19
Poly (N-isopropylacrylamide)	Hydrophilicity & underwater Superoleophobicity/ Hydrophobicity & superoleophilicity	√	O/W W/O	-	-	-	16
Fluoropolymer/SiO ₂ , stainless steel mesh	superhydrophobic/ Superoleophilic	√	W/O	-	-	-	38
Aluminum phosphate, TiO ₂	Superamphiphilic/ Underwater superoleophobic	√	O/W	-	200 L m ⁻² h ⁻¹	√	31
Poly (melamine formaldehyde), SiO ₂	Superhydrophilic/ Underwater superoleophobic	√	O/W	-	2.5 × 10 ⁵ L m ⁻² h ⁻¹ bar ⁻¹	√	41
GO, C ₃ N ₄ , TiO ₂	Superhydrophilic/ Underwater Superoleophobic	√	O/W	√	4536 L m ⁻² h ⁻¹ bar ⁻¹	√	51
Acrylic acid and styrene/ divinyl benzene	Hydrophilic/ Oleophilic	√	O/W	√	-	-	43
Hydrolyzed polyacrylonitrile	Superhydrophilic/ Underwater superoleophobic	√	O/W	-	5152 L m ⁻² h ⁻¹	√	42
Melamine sponge	Superhydrophobic/ Superoleophilic	-	W/O	-	156 700 L m ⁻² h ⁻¹ bar ⁻¹	-	40
Divinylbenzene, PVDF	Superhydrophobic/ Superoleophilic	-	W/O	√	1500 L m ⁻² h ⁻¹	-	52
Attapulgite, PVDF	Superhydrophilic/ Underwater Superoleophobic	-	O/W	-	360 L m ⁻² h ⁻¹ bar ⁻¹	-	32
Photocurable resin PDMS	Superhydrophobic/Superoleophilic/ Under-water superoleophobic/ Under-oil superhydrophobic	√	O/W	-	> 2000 L m ⁻² h ⁻¹	√	This work

15. How was the effective surface area calculated for the flux measurements?

Response: Thanks for the referee's comment and question. The numerical results of the separation flow were calculated according to the formula on Page 16, line 352, $J = 2V_c/HWt$. Here, we have learned from the calculation method of oil and water separation experiment by superhydrophobic meshes. We take the cross-section area where the liquid passes through the joint of the two gears as the effective surface area. As shown in Figure R18, HW product is our definition of effective surface area.

Figure R17. Micro-CT imaging of cross-sectional views of two gears.

16. How can this device be used in practical oil spill cleanup operation in open waters?

Response: Thanks for the referee’s comments. Our dual-bionic superwetting gears device has two advantages for oil spill cleanup operations in open waters: **First**, open sea oil pollution is a system of mixed oil and water in various scales. Our device can solve this problem. **Second**, the effective anti-fouling ability of our dual-bionic superwetting gears device can ensure that the separation flow does not decrease for a long time, which has practical use potential. We also believe further engineering efforts are needed to design more effective oil spill cleanup operations in open waters. For example, Figure R18 shows one potential solution of using a pump to drip contaminated seawater over our dual-bionic superwetting gears device to separate water and oil on a boat, 1.0 m long and 0.8 m wide, in a swimming pool (2.2 X 1.5 X 0.5 m).

Figure R18(Extended Data Fig.9). Dual-bionic superwetting gears device separating water and oil in a swimming pool. a, Schematic of the oil leaked in open waters. **b,** Optical images of the boat-integrated dual-bionic gears separator. **c,** Time sequences of the separation process for gears device to separate water and oil.

Our modification to the manuscript: On page 17, lines 372-377, we added described of the potential solution of a dual-bionic superwetting gears device to separate water and oil in open waters: “Next, a proof-of-concept experiment is performed in a swimming pool with a length of 2.2 m, a width of 1.5 m and a water depth of 0.5 m to simulate the cleaning ability in the oil spill accident. The device is mounted on a boat of 1.0 m long and 0.8 m wide, where a pump driven by the solar cell drips contaminated seawater onto the gears (Fig. 4e, Extended Data Fig. 9). The effective ability of our dual-bionic superwetting gears device can ensure that the separation flow does not decrease for 240 minutes.”

17. Please add the missing scale bars in Extended Data Fig. 5.

Response: Thanks for the referee’s comments. We add the emulsion dispersing phase’s particle size in Figure 4b. The distribution (Figure R12) were analyzed by the laser granularity instrument (Winner319C).

Our modification to the manuscript: On page 16, lines 329-331, we added described the particle size of the emulsion: “In addition, the emulsion dispersing phase’s particle size and distribution (Fig. 4b) were analyzed by the laser granularity instrument (Winner319C) and the average size of the oil droplets is $\sim 100 \mu\text{m}$.”

Figure R12 The particle size of the emulsion. a,c, Schematic of emulsion before separation (a) and after separation (c). b, Histogram of emulsion particle size distribution. d, e, Optical and stereomicroscopic images of collected water (d) and collected oil (e).

18. The authors can use a simpler and more straightforward language through the text to better convey their message to the readers. Example (Title): “co-design” or “design”? what is “multi-scaled” referring to? Please revise.

Response: Thanks for the referee’s comments. “Design” refers to a participatory approach to designing structures, in which the designed agricultures are treated as equal collaborators in the design process. In our system, the dual-bionic superwetting gears, including the combination of the cat-tongue biomimetic structures and *Nepenthes*-pitcher-peristome cavities, could form complementary topological structures that can match together. The ‘multi-scaled’ means that the bionic gears can separate oil-water mixtures in different scales, for example, oil-water micro-drops on a millimeter scale, and oil-in-water emulsions on the micrometer scale.

Based on the referee’s suggestion, we replaced the word ‘co-design’ with ‘design’. A simpler and more straightforward language can better convey the message to the readers. Thanks again for the referee’s suggestion.

REVIEWERS' COMMENTS

Reviewer #1 (Remarks to the Author):

Report on **Designing of Dual-Bionic Superwetting Gears with Liquid Directional Steering for More Sustainable Multi-Scaled Oil-Water Separation**

by Zhuoxing Liu et al.,

Submitted to *Nature Communications*

Comments for the authors

I still have some concerns about the authors' reply. Please reconsider the comments as follows.

1. In comment 2, we commented on the validity of the benchmark analysis in fabrication time and device strength. However, the analytical method in fabrication time was based on the author's estimation, and the device strength was not fair.

2. In comment 5, we commented on the validity of the antifouling performance of the device. The authors exhibited a low degradation in separation flux; however, this data is not directly linked to the antifouling performance. In the revised version, antifouling performance is confirmed from the detachment of the single millimetric bead from the device; however, this is not enough. It is apparent that the millimetric bead is removed by gravity. If small wettable solids (e.g., sand) are attached to the device, the lubrication layer must adsorb them. I agree that the device exhibits long-term performance stability.

3. In comment 7, we commented on the possibility of a high-oil contents mixture can be separated. The authors agreed with the separation of high-oil contents mixture; however, it is not evidenced.

4. Extended Data Fig. 9 in the revised manuscript is excellent. Since this work focuses on the device design, I recommend that this data be in the main figure.

Reviewer #2 (Remarks to the Author):

The authors demonstrated the continuous separation with the rotating pre-wetted gears to separate oil and water for water purification. In addition, It is confirmed that the authors had an effort to explain the mechanism of sequence separation procedures in detail. The authors tried to show the anti-fouling performance of using solid contaminants. Therefore, this work can expand scientists' views in the field of oil/water separation and is recommended for publication in Nature Communications without additional revision. As further work, I recommend experiments related to the continuous separation of mixed forms (solid, emulsion, and oil/water).

Reviewer #3 (Remarks to the Author):

I am satisfied with the revision made by the authors, and I am happy to recommend the revised manuscript for publication in Nature Communications in its current form.

Reviewer #4 (Remarks to the Author):

The authors have made substantial improvement to the submitted manuscript. I believe it is now ready to be revealed to the scientific community. Many thanks and congratulations to the authors for this great contribution and hopefully this innovative device will have the expected impact in the field of oil-water separation and oil spill cleanup. It is expected to move this field forward towards finding practical solutions for such challenging environmental issue.

Point-by-point response

Reviewer comments are set on grey background, the responses to the comments and newly added text in the revised manuscript and SI are shown in blue.

Responses to Referee # 1

Referees' Comments:

Comments for the authors

I still have some concerns about the authors' reply. Please reconsider the comments as follows.

1. In comment 2, we commented on the validity of the benchmark analysis in fabrication time and device strength. However, the analytical method in fabrication time was based on the author's estimation, and the device strength was not fair.

Response: Thanks a lot for the referee's comments and questions. The statistics of fabrication time and device strength are provided by the relevant research. In terms of fabrication time, there may be a slight deviation in time assessment, but it will not affect the difference in time required for different materials. For example, 3D multiscale superhydrophilic sponges in reference 45 require at least 100h while the attapulgate coated membranes in reference 36 can be completed within 2h. The fabrication time and strength are critical and meaningful parameters for oil-water separation materials and have a strong reference value for the practical application of oil-water separation materials. To demonstrate the benchmark analysis, we replaced the color bar by defining the point with certain color with a fabrication time according to the reference in the revised manuscript.

2. In comment 5, we commented on the validity of the antifouling performance of the device. The authors exhibited a low degradation in separation flux; however, this data is not directly linked to the antifouling performance. In the revised version, antifouling performance is confirmed from the detachment of the single millimetric bead from the device; however, this is not enough. It is apparent that the millimetric bead is removed by gravity. If small wettable solids (e.g., sand) are attached to the device, the lubrication layer must adsorb them. I agree that the device exhibits long-term performance stability.

Response: Thanks a lot for the referee's comments. In our experiment, our dual-bionic superwetting gears device is in a constant state of superwettability, underwater superoleophobic, underoil superhydrophobic materials can prevent the adhesion of liquid oil or water phase that showed a very good anti-fouling performance. Many thanks for the

referee's valuable suggestions and we will take your comments into account in our future research.

3. In comment 7, we commented on the possibility of a high-oil contents mixture can be separated. The authors agreed with the separation of high-oil contents mixture; however, it is not evidenced.

Response: Thanks a lot for the referee's meaningful comments. During the experiment, the high-oil contents mixture can also be separated. Surface superwettability and complementary topological structures are essential for our device, the water and oil in the mixture can steer to spread on the preferential gear surfaces rapidly and continuously, forming the respective liquid films to repel the other liquid. Even if the oil content is increased, the device can separate well.

4. Extended Data Fig. 9 in the revised manuscript is excellent. Since this work focuses on the device design, I recommend that this data be in the main figure.

Response: Thanks a lot for the referee's comments. We have moved the Supplementary Figure 9 to the Figure 4.

Responses to Referee # 2

Referees' Comments:

The authors demonstrated the continuous separation with the rotating pre-wetted gears to separate oil and water for water purification. In addition, it is confirmed that the authors had an effort to explain the mechanism of sequence separation procedures in detail. The authors tried to show the anti-fouling performance of using solid contaminants. Therefore, this work can expand scientists' views in the field of oil/water separation and is recommended for publication in Nature Communications without additional revision. As further work, I recommend experiments related to the continuous separation of mixed forms (solid, emulsion, and oil/water).

Response: We thank the referee for the positive comments and insightful suggestions to improve the quality of our manuscript. We will take your suggestions into account in our future work.

Responses to Referee # 3

Referees' Comments:

I am satisfied with the revision made by the authors, and I am happy to recommend the revised manuscript for publication in Nature Communications in its current form.

Response: We thank the reviewer for the invaluable comments and feedback.

Responses to Referee # 4

Referees' Comments:

The authors have made substantial improvement to the submitted manuscript. I believe it is now ready to be revealed to the scientific community. Many thanks and congratulations to the authors for this great contribution and hopefully this innovative device will have the expected impact in the field of oil-water separation and oil spill cleanup. It is expected to move this field forward towards finding practical solutions for such challenging environmental issue.

Response: We thank the reviewer for the recognition and support of our work.